# BEDD: The MineRL BASALT Evaluation and Demonstrations Dataset for Training and Benchmarking Agents that Solve Fuzzy Tasks

**Stephanie Milani**
Carnegie Mellon University
smilani@cs.cmu.edu

Anssi Kanervisto
Microsoft Research
anssi.kanervisto@microsoft.com

Karolis Ramanauskas
University of Bath
kr711@bath.ac.uk

Sander Schulhoff
University of Maryland
sschulho@umd.edu

Brandon Houghton
OpenAI
brandon@openai.com

Rohin Shah
rohinmshah@gmail.com

## Abstract

The MineRL BASALT competition has served to catalyze advances in learning from human feedback through four hard-to-specify tasks in Minecraft, such as *create and photograph a waterfall*. Given the completion of two years of BASALT competitions, we offer to the community a formalized benchmark through the BASALT Evaluation and Demonstrations Dataset (BEDD), which serves as a resource for algorithm development and performance assessment. BEDD consists of a collection of 26 million image-action pairs from nearly 14,000 videos of human players completing the BASALT tasks in Minecraft. It also includes over 3,000 dense pairwise human evaluations of human and algorithmic agents. These comparisons serve as a fixed, preliminary leaderboard for evaluating newly-developed algorithms. To enable this comparison, we present a streamlined codebase for benchmarking new algorithms against the leaderboard. In addition to presenting these datasets, we conduct a detailed analysis of the data from both datasets to guide algorithm development and evaluation. The released code and data are available at https://github.com/minerllabs/basalt-benchmark.

## 1 Introduction

In traditional reinforcement learning, an agent learns how to act using reward based on an explicitly-defined reward signal [42]. This reward signal is often carefully designed by domain experts to communicate the intended goal for the agent to accomplish. Precisely specifying this form of feedback programmatically requires designers to *a priori* enumerate all potential outcomes or constraints on how they would like the task to be completed. This enumeration is difficult to achieve in practice, and the resulting reward signals often fall short at correctly specifying the designer's intent [29]. To address this challenge, researchers have explored the idea of incorporating alternative channels for communicating information about the desired behavior of the agent. This class of techniques is generally called *learning from human feedback* (LfHF) [8, 23]. The goal of LfHF is to utilize the feedback modalities most likely to result in an agent acting according to human-desired specifications.

37th Conference on Neural Information Processing Systems (NeurIPS 2023) Track on Datasets and Benchmarks.

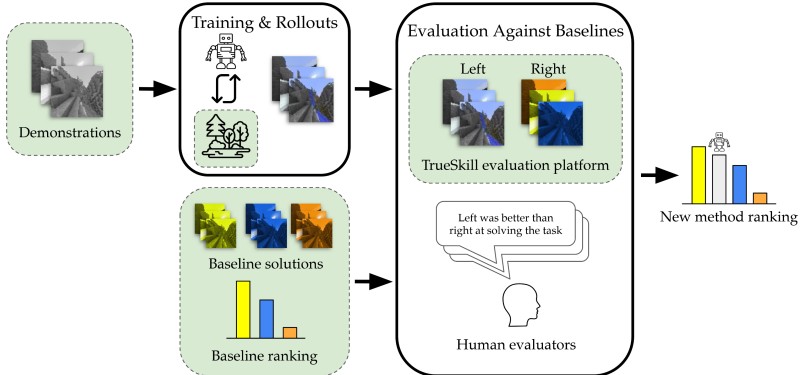

Figure 1: The BASALT benchmark. Green parts show the components of the included datasets. The agents are compared using video recordings of them solving the task. We contribute data for training agents to perform fuzzy tasks and for evaluating agents with human evaluators. We contribute code for further benchmarking.

The complexity of this approach has been exemplified in the MineRL BASALT competitions [39], which leverage the open-ended and flexible Minecraft environment to promote the development of LfHF algorithms. This competition series has provided a platform for developing agents capable of solving fuzzy tasks that lack well-defined reward signals. However, despite two years of competition, no agents have yet matched human performance levels [33], emphasizing the need to establish the BASALT tasks as a consistent, standardized benchmark. In this paper, we clarify the BASALT benchmark and present concrete evaluation recommendations toward the goal of consistency.

To support this benchmark we introduce the BASALT Evaluation and Demonstrations Dataset (BEDD), an open and accessible dataset for learning to solve fuzzy tasks from human feedback. As shown in Figure 1, this dataset consists of three main ingredients: the `Demonstrations Dataset`, the `Evaluation Dataset`, and supporting code for utilizing and analyzing the data. The `Demonstrations Dataset` consists of over 26 million image-action pairs from 14,000 videos of labeled Minecraft gameplay of human players completing the BASALT tasks. To facilitate evaluating agents using real human judgments, we present the `Evaluation Dataset`, derived from the most recent BASALT competition [25]. The `Evaluation Dataset` consists of over 3,000 dense pairwise human evaluations of videos of various agents performing the BASALT tasks. By dense, we mean that each evaluation includes a comparison of the relative task-completion performance of the agents and at least four additional questions, such as which agent was more human-like. This results in 27,905 comparison points between 17 different agents. This dataset also includes a natural language response justifying why an agent was selected as being the better of the two. The provided data functions as a leaderboard, offering researchers the ability to compare their newly-developed algorithm against various agents without redoing all costly human evaluations from scratch.

To facilitate the use of the `Demonstrations Dataset` and the `Evaluation Dataset`, we present a streamlined codebase. With this codebase, one can train a new model from the demonstration dataset and evaluate it against the provided leaderboard. Alongside the presentation of these datasets, we conduct a detailed analysis of the data to guide algorithm development and evaluation. We release the code and detailed documentation for others to perform such analyses. We hope that this codebase will assist others with quickly developing and evaluating LfHF algorithms to spur further progress toward agents that are better aligned with human intent.

## 2  The MineRL BASALT Benchmark

We provide an overview of the MineRL BASALT benchmark, consisting of a task suite and evaluation framework for learning from human feedback, illustrated in (Figure 1). The benchmark uses Minecraft, a videogame that provides a rich and complex environment in which to define different tasks. The states are pixel observations; the actions are regular keyboard and mouse actions, closely following how humans play the game. This includes navigating the crafting menus using a mouse (Figure 2).

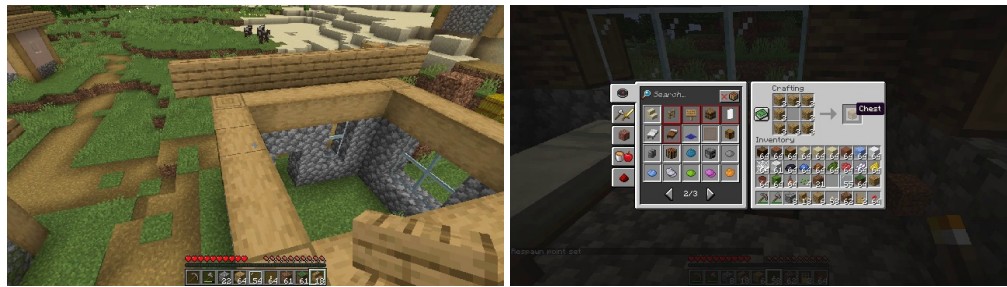

Figure 2: Two example images of the pixel observations provided by the MineRL environment used by BASALT. The agent receives pixels as observations and must use regular keyboard and mouse actions, including navigating the crafting menu with a mouse to make items.

## 2.1 Tasks

All tasks are accompanied by a Gym environment [4] and either a simple English-language task description or a reward function to indicate the desired agent behavior. The *reward-free* tasks have the language task specification; the *rewarded* tasks are accompanied by a reward function.

We use the same four reward-free tasks from the BASALT competitions, since these tasks were empirically validated to be challenging [33, 38]. The goal of FindCave is for the agent to discover a naturally-generated cave. To complete MakeWaterfall, the agent must create a waterfall and subsequently take a picture of it. Because taking a photograph is not supported in Minecraft, we simulate it by providing the agent with an option to end the episode: the state at the moment that the agent ends the episode is its photograph. In CreateVillageAnimalPen, the agent must build an animal pen next to an existing house, then corral a matching pair of farm animals into the pen. In BuildVillageHouse, the agent spawns in a village and must build a new house in the same style as the other surrounding houses – without damaging the village. For more task details, see Appendix A.

Because reward-free tasks are challenging to evaluate, we include ObtainDiamondShovel [33], a task with a concrete reward function, to enable quick iteration. This task is more challenging than ObtainDiamond [17], a well-established benchmark in reinforcement learning, since it requires an additional crafting step after obtaining the diamond. An agent receives reward each time it obtains the next required resource in the crafting tree towards a diamond shovel.

## 2.2 Evaluation

To fairly compare algorithms on a benchmark, the parameters of that benchmark must be concretely defined. Previously, many techniques claimed to solve MineRL ObtainDiamond. Often, these successes involved giving the agent access to the full game state instead of pixel observations, building in human priors through extensive action shaping, or simplifying the underlying environment dynamics. Changes like these should be clearly stated and applied equally to compared methods. To avoid such discrepancies in the future, we specify some methodological best practices when using the BASALT benchmark: i) only pixel observations provided by the four environments should be used, ii) the environments should not be modified in any way, iii) action shaping [26] is permitted, as long as it applied to all methods, iv) algorithms should be evaluated with TrueSkill [20] using the specific hold-out test seeds provided in Appendix A, and v) the final evaluations must be conducted with human evaluators. For further guidance on how to fairly compare algorithms, we refer the reader to recent work [35].

BASALT agents are evaluated on hold-out test environments to produce multiple videos of agents attempting to complete that task. A single trial consists of two agents pitted against one another, which is shown to a human judge. This judge determines which agent completed the task better. The resulting dataset of human evaluations is assessed using the TrueSkill system, which dynamically assesses the skills of a particular player (in this case, an LfHF algorithm). In addition to estimating the relative skill of an algorithm, it provides an uncertainty estimation. For a more extensive discussion of this procedure and a comparison of TrueSkill to other rating systems, please see Appendix A. We hope that our demonstrations of successful task completions and the benchmarked algorithms from the competition can serve as a starting point for evaluating the performance of other algorithms.

Developing algorithms while leveraging real human feedback is both expensive for machine learning practitioners and time-consuming for human evaluators. Automated human evaluations have emerged as a valuable part of the pipeline for assessing various aspects of machine learning models [7, 10]. The goal is not to replace human evaluations; instead, it complements the process by providing quick, iterative feedback for algorithm development and initial assessment. As a result, we propose *automating evaluations* [12] as an additional component of the BASALT benchmark. The inclusion of ObtainDiamondShovel may help develop reward modeling techniques [22] due to its associated concrete reward function. More generally, we hope that with the release of the Evaluation Dataset, others can begin developing approaches toward this goal.

## 2.3 Benchmarking Algorithms on BASALT

To assist with the development of LfHF algorithms and automated evaluations, we implement and share a codebase with two major contributions. First, the code contains an example of training a LfHF algorithm with the shared data in the Demonstrations Dataset. Second, we include the tools for performing the evaluations presented in this work. The code is a Python-installable library, which allows the functionality to be imported into other codebases for use in research.

The training example provides tools to train a behavior cloning model on top of the Video PreTraining (VPT) [2] model using the imitation [14] library. VPT is a large foundation model that can complete various tasks in Minecraft, but it is difficult to fine-tune on new tasks due to its size. Inspired by the success of an imitation-learning approach[1] in the BASALT 2022 competition, we use behavior cloning as the base algorithm with the rich embeddings from the VPT model as input. Motivated by the original VPT results, we remove the no-op actions from the demonstration dataset. We provide code to use different variants of the VPT model for benchmarking. The final output is a set of videos to use in evaluations against the shared recordings of other agents in the Evaluation Dataset.

Because setting up human evaluations is time-consuming, we share our platform for conducting human evaluations. The platform is served as a webpage backed by a simple Python-based webserver. The collected data is either stored locally in a SQLite database or remotely in a more scalable form. The platform includes a flexible API to add or remove agents from the set of comparisons. After a simple setup, researchers can point human evaluators to a specific URL to provide answers. We provide examples of the form that the human evaluators will see in Appendix D. We also share code to create figures of the TrueSkill ratings (and all analyses in Section 4), given the answers contained in the resulting database. This whole process serves as an end-to-end example of creating a new method with Demonstrations Dataset, then evaluating it with data from Evaluation Dataset.

# 3 BASALT Evaluation and Demonstrations Dataset

We now introduce BEDD, our extensive dataset of human demonstrations and algorithm evaluations. This dataset consists of the following components:

- The Demonstrations Dataset, a set of 13,928 videos (state-action pairs) demonstrating largely successful task completion attempts of the reward-free tasks,
- The Evaluation Dataset, a set of 3,049 dense pairwise comparisons of algorithmic and human agents attempting to complete the BASALT tasks, and
- The code for utilizing and analyzing these datasets for developing LfHF algorithms (some details in Section 2.3).

For a full datasheet [13], please see Appendix B.

## 3.1 Demonstrations Dataset: Completing Reward-Free Tasks in Minecraft

The Demonstrations Dataset for developing new methods consists of 361 hours (26 million image-action pairs) of human demonstrations of the reward-free BASALT tasks. This data consists of labeled trajectories, both with high-resolution image observations and keyboard and mouse actions for each frame. In total, this is 651 GB of data. Table 1 decomposes the high-level data statistics by task. More details about this dataset are in Appendix C.

---

[1] https://github.com/shuishida/minerl_2022

| Task | Videos | Episodes | Hours | Size | Ep. len, s | Success % |
|------|--------|----------|-------|------|-----------|-----------|
| FindCave | 5,466 | 5,466 | 91 | 165GB | 60 | 93% |
| MakeWaterfall | 4,230 | 4,176 | 97 | 175GB | 84 | 98% |
| CreateVillageAnimalPen | 2,833 | 2,708 | 89 | 165GB | 119 | 95% |
| BuildVillageHouse | 1,399 | 778 | 85 | 146GB | 391 | 92% |
| Total | 13,928 | 13,128 | 361 | 651GB | 99 | 95% |

Table 1: High-level demonstration data statistics decomposed by task. Episode length is the average episode length in seconds. A demonstration is counted as success if the player manually ended the episode instead of dying or timing-out.

| Task | Comparisons | Hours | Words in Response | Response Sentiment 👍 | 👎 | 🖐 |
|------|-------------|-------|-------------------|------------------|-----|-----|
| FindCave | 722 | 60 | 27,948 | 79% | 14% | 7% |
| MakeWaterfall | 682 | 56 | 26,437 | 76% | 7% | 17% |
| CreateVillageAnimalPen | 914 | 81 | 32,768 | 57% | 11% | 32% |
| BuildVillageHouse | 731 | 76 | 26,917 | 63% | 9% | 28% |
| Total | 3,049 | 273 | 114,070 | | | |

Table 2: High-level evaluation data statistics decomposed by task. We report the total number of agent-agent comparisons, human labor hours, and words used in the natural-language justifications of selecting a specific agent as the best one. We also report the percent of positive, neutral, and negative sentiments in these justifications.

Each demonstration consists of a trajectory $\tau = [s_0, a_0, \ldots, s_N, a_N]$, or a sequence of state-action pairs, where $N$ is the trajectory length. These pairs are contiguously sampled at every Minecraft game tick (20Hz). Each state consists of the 640x360 RGB frame from the perspective of the player (see Figure 2). Each action consists of two parts $a = [K, M]$, where $K$ is all keyboard interactions, $M$ is all "mouse" interactions (change in view, pitch, and yaw), mimicking the native human control interface of Minecraft. This dataset serves as a starting point for using *demonstrations* as a form of feedback to train agents.

### 3.2 `Evaluation Dataset`: **Evaluating BASALT Agents**

The `Evaluation Dataset` contains 3,049 pairwise comparisons of different algorithms, produced from 273 hours of human labeling effort by 65 unique MTurk workers. All responses are contained in a single JSON file. Table 2 decomposes this dataset by task. Each evaluation in the dataset consists of the following: (i) the names of the two agents used in the comparison, (ii) the corresponding videos shown to the human judge, (iii) an answer to the question of which player is better overall (Left, Right, Draw), (iv) a natural-language justification of this choice, (v) answers to at least one direct question about concrete achievements by the players (Left, Right, or Both), and answers to 4-7 comparative questions, such as which agent was more human-like (Left, Right, Draw, N/A). The direct and comparative questions are task-specific. Including the choice of which algorithm performed best, this dataset consists of a total of 27,905 comparisons along various factors.

We provide all responses in our public release of the data. We also include a list of anonymized MTurk workers whose responses we found not to suit our standards (e.g., providing the same answer to every task). Before performing the analyses reported in this paper, we filtered the data to exclude these responses. We provide all details needed to understand and reproduce these evaluations in Appendix D. With this dataset, one may compare a newly-developed algorithm with 17 possible agents: the top 13 teams in the 2022 BASALT competition, a behavioral cloning baseline, a random agent, and two human experts (two of the authors of this paper). Researchers can use the provided human evaluations to kickstart the human evaluation, without needing to dedicate costly human labor to evaluating all algorithms from scratch.

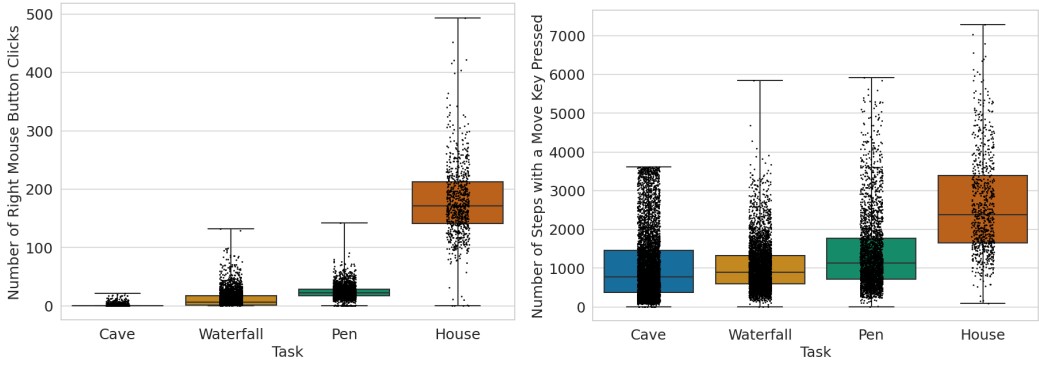

(a) Right mouse button clicks          (b) Movement key presses

Figure 3: The distributions of the number of right mouse button clicks and movement key presses across tasks in the `Demonstrations Dataset`, which act as proxies for the number of blocks placed and distance traveled per episode, respectively. Because `BuildVillageHouse` takes the longest to complete, it is reasonable that it requires more right mouse button clicks and movement key presses.

## 4   Analysis: `Demonstrations Dataset`

We now analyze the `Demonstrations Dataset`. Because the BASALT tasks lack concrete reward functions, evaluating the progress of agents on these tasks is challenging. This difficulty necessitates the use of informative proxy measures. As a result, when analyzing the `Demonstrations Dataset`, we focus on defining proxy measures that may be useful for understanding the data or tracking training progress. In this section, we describe these proxies and present the results of our analysis.

We first seek to understand the relative difficulty of the four reward-free tasks. Given the experience of the contracted data collectors with Minecraft, we believe that the length of the demonstration is a reasonable proxy for task difficulty. Using demonstration length as a proxy for task difficulty, we note that `BuildVillageHouse` is likely the most challenging task, even for humans: each video lasts around 6.52 minutes on average, while the next most time-consuming task, `CreateVillageAnimalPen`, takes an average of around 1.98 minutes (Table 1). By this metric, the easiest task is `FindCave` (1 minute). In practice, one could use this metric to assess training progress or likelihood of task completion. If an agent completes a task at a rate that is far away from the average, that may signal worse quality behavior. In contrast, if an agent takes a similar amount of time to the average to complete a task, then that could signal better behavior (but is clearly not definitive).

We also want to understand when an agent — human or AI — may be underperforming on the task. Because `FindCave` requires navigation to find a cave, an agent that remains stationary is likely unsuccessful. Similarly, because `CreateVillageAnimalPen` and `BuildVillageHouse` require the construction of objects, agents that fail to place any blocks likely do not succeed at the task. As a result, we employ the number of steps with an active movement key as a proxy for the distance traveled within the tasks. We also use the count of right mouse button clicks as a proxy for the number of blocks placed. The only other actions performed by clicking the right mouse button are using a crafting table or a chest, of which there are relatively few per episode.

These proxies are depicted in the per-episode distributions within the `Demonstrations Dataset` in Figure 3. The number of right mouse button clicks is highest at around 180 in `BuildVillageHouse` and between 20-40 in `MakeWaterfall` and `CreateVillageAnimalPen`. These align with expectations, given the amount of building required to complete the tasks. The number of movement actions has the highest variability in `FindCave`. This task also has the shortest time-out at 3,600 steps. Perhaps a better proxy would be the average number of movement actions per step: `FindCave` should have the highest number, as the task mainly consists of moving around.

We include these results both as a way to understand this dataset and as values to monitor during agent training to estimate agent performance. However, it is crucial to avoid using these metrics as direct optimization targets, as they can be easily exploited.

| Agent | Normalized TrueSkill | Sentiment 👍 | 👎 | 🖐 |
|---|---|---|---|---|
| Human2 | 2.43 | 92% | 5% | 3% |
| Human1 | 2.17 | 92% | 6% | 2% |
| GoUp | 0.73 | 74% | 19% | 7% |
| UniTeam | 0.13 | 66% | 25% | 9% |
| BC-Baseline | −0.32 | 65% | 26% | 9% |
| Random | −1.35 | 63% | 29% | 8% |

Table 3: Normalized TrueSkill score and percent of positive, neutral, and negative sentiments of the natural language justifications for selecting an agent as the best one.

## 5 Analysis: `Evaluation Dataset`

We first analyze the overall dataset, then focus on a few main comparisons: the top two algorithms from the BASALT 2022 competition (GoUp and UniTeam), the behavioral cloning baseline (BC-Baseline), the two human experts (Human1 and Human2), and a random agent (Random). This subset of data corresponds to 394 of 3,049 total comparisons or nearly 34 hours of human labeling effort. For additional details about the subsequent analyses, please see Appendix E.

### 5.1 Task-Based Analysis

Table 2 provides a high-level overview of the 3,049 total evaluations, decomposed by task. The human judges generally responded using a similar number of words across the different tasks. Only the responses to `FindCave` and `MakeWaterfall` contained significantly more words on average than `CreateVillageAnimalPen` ($F = 3.72$, $p = 0.01$; 95% CI: 0.34 to 5.37, $p = 0.02$; 95% CI: 0.36 to 5.47, $p = 0.02$, respectively).[2] All other tasks exhibited no significant differences between means. This result suggests that expressing the specific rationale for these simpler tasks may be easier.

To understand the general perceptions of the human judges of the different tasks, we categorized each response into positive, neutral, or negative sentiment [31]. We then analyzed the differences in the distribution across tasks. The human judges displayed different levels of sentiment in their responses, depending on the task.[3] They responded most positively to `FindCave` and least positively to `CreateVillageAnimalPen`. The only sentiment distributions that were *not* significant were between `AnimalPen` and `BuildVillageHouse` ($X^2(1, N = 1645) = 6.35, p = 0.042$), and `FindCave` and `MakeWaterfall` ($X^2(1, N = 1404) = 2.694, p = 0.260$). All other pairs of tasks exhibited significant differences in sentiment.[4] Due to the large number of subtask dependencies required for completing `CreateVillageAnimalPen` and `BuildVillageHouse` (as well as the greater amount of time required on average for human demonstrators to complete these two tasks, detailed in Table 1 and Section 4), they are considered to be more challenging than `FindCave` and `MakeWaterfall`. The more positive sentiment toward `FindCave` and `MakeWaterfall` may be due to an increased focus on successes due to their relative ease of completion.

### 5.2 Agent-Based Analysis

We now turn our attention to agent-based analysis. We first provide a brief overview of the included agents. GoUp uses human knowledge to decompose the tasks into the same high-level sequence, then uses computer vision techniques to identify the goal for each task (e.g., the cave). UniTeam combines behavioral cloning with search by embedding the current set of images with a pre-trained VPT network and then searching for the nearest embedding point in the VPT latent space to find the situation to use as reference and copying the corresponding expert actions [32]. More details about the other assessed agents can be found in previous work [33].

---

[2]We conducted a one-way ANOVA with a post-hoc Tukey's HSD test to obtain these results. Results presented as (ANOVA $F$ value, $p$ value; CI for Tukey's HSD, $p$ value).

[3]We conducted a chi-square test of independence to examine the relationship between task and sentiment classification. The relation between these variables was significant, $X^2$ (6, $N = 3049$) = 132.21, $p < .001$.

[4]These results are from Bonferroni-corrected pairwise Chi-square tests.

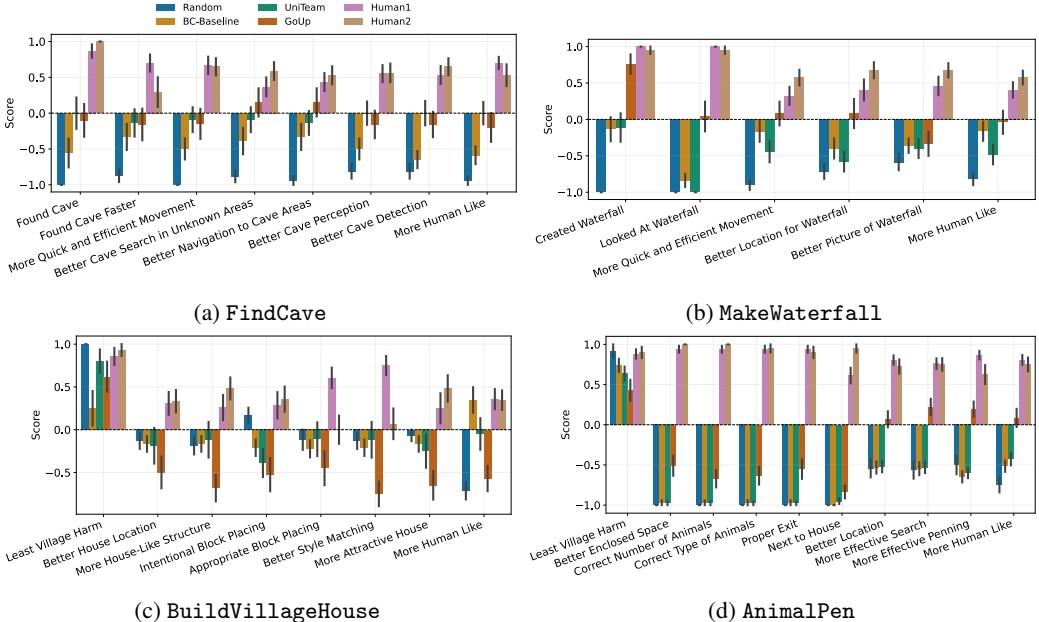

(a) `FindCave`

(b) `MakeWaterfall`

(c) `BuildVillageHouse`

(d) `AnimalPen`

Figure 4: Comparison of baseline solutions, top BASALT 2022 competition solutions, and humans on additional questions. Bars represent average score, and error bars represent standard error. Higher is better: it means that agent exhibited more of that factor according to human judges.

We present the agent-based overview of the evaluations in Table 3. Note that the reported TrueSkill scores are computed over the 3,049 total comparisons, whereas the sentiment analysis is performed only on the 394 agent-specific entries. The human agents performed the best using TrueSkill ranking, with a large performance gap between them and the best-performing algorithmic agent. This result emphasizes the difficulty of BASALT for algorithmic agents.

The only sentiment distributions that *were* significant were between either of the two human experts and the other algorithms. In particular, the sentiment toward comparisons that included human players were significantly more positive.[4] All other pairs of algorithms did not exhibit significant differences in sentiment. In general, human judges were generally positive when evaluating the agents. The judges may have been more positive due to social influence: they knew that the videos were produced by different players for ranking in the competition.

We now investigate the responses to the direct and comparative questions. For succinctness, we summarize each question into a single factor. For example, we use "More Human Like" to capture the question, "Which player seemed more human-like (rather than a bot or computer player)?". Figure 4 presents these results. Importantly, no machine learning algorithms outperform humans on any of the factors, indicating that there is still plenty of improvement to be made. When analyzing future algorithms, we suggest preserving this decomposition to both validate that the responses are sensible and obtain a fine-grained understanding of agent capabilities. To validate sensible responses, we suggest checking the scores for the *random* agent: except for causing the least harm, this agent should score poorly compared to the other agents.

Finally, we highlight a few interesting findings that are enabled by our decomposed and extensive set of evaluation criteria. Figure 4b (`MakeWaterfall`) in the paper reveals that, although Team GoUp's algorithm can create waterfalls at a rate more similar to the human players, it struggles along all other criteria, including choosing a good location and taking a high-quality photograph. Only looking at the binary success/failure condition for creating a waterfall would ignore these important nuances in the algorithm's behavior. As another example, Figure 4a (`FindCave`) suggests that, while Team GoUp's algorithm still struggles to find caves, it can reasonably search for and navigate to areas that are likely to have caves. This finding suggests that the performance bottleneck may be the cave detection system employed by this approach.

## 6 Related Work

**Learning from Human Feedback**  Learning from human feedback has become a crucial research direction in machine learning [1, 8, 39], aiming to leverage human expertise to improve algorithm performance and generalization. Various algorithms have been proposed to integrate human feedback into the learning process, often using techniques such as imitation learning [21], inverse reinforcement learning [5, 45], and reward modeling [41]. Most commonly, these algorithms are evaluated in standard reinforcement learning benchmarks [3, 9, 11, 43]. However, most of these benchmarks do not have the property that human feedback is *critical* for task identification, as in BASALT. In some Atari games, if an agent does anything other than the intended gameplay, it dies and resets to the initial state, so pure curiosity-based agents perform well [6]. In contrast, BASALT tasks require human feedback or data to identify and complete tasks.

**Minecraft for Machine Learning**  Minecraft is as a valuable platform for machine learning research [19, 30, 40] due to its complex, dynamic environment and customizable game engine. As a result, numerous competitions and benchmarks [16, 18, 24, 37] have been developed to improve AI capabilities. These often include tasks with concrete or programmatic reward functions, such as learning to collect diamonds in a sample-efficient way [17], multi-agent learning of cooperative and competitive tasks [36], and more [12]. In contrast, we emphasize tasks that are designed to be hard to specify through a reward signal. Other work focuses on natural language as a specific modality for communicating intent [15, 27, 28, 34]; instead, we aim to promote the development of algorithms that generally learn from human feedback, including natural language. Some techniques require complete game state information (knowledge of blocks and items around them) [44, 46]; however, our benchmark only permits agents to access observations, not game state, to promote learning from pixels, which is more generalizable to other tasks.

**Comparison to MineDojo**  Perhaps the most similar benchmark to ours is MineDojo [12], which emerged during the BASALT competitions and contributed a large dataset of Minecraft gameplay with the aim of developing generally-capable embodied agents. The focus of MineDojo was to provide a massive dataset scraped from the internet; in contrast, we focus on curating a smaller set of high-quality demonstrations and evaluations. Our demonstration data was produced by experienced Minecraft data collectors using a consistent Minecraft version and settings. In contrast, the MineDojo data, although plentiful, contains many videos with streamer overlays and non-Minecraft parts, different texture packs, mod packs, and more. Although this diversity may be beneficial in some settings, previous work had to filter out such data before it could be useful for training [2]. Another critical difference is evaluation: MineDojo provides only binary success or failure criterion; instead, our recommended evaluation and resulting `Evaluation Dataset` involves human judgments across a range of quantitative (e.g., Found Cave) and qualitative (e.g., Style Matching) criteria.

## 7 Conclusion

We proposed BEDD, a large and accessible dataset to facilitate algorithm development for the BASALT benchmark on learning from human feedback. The benchmark consists of five tasks in Minecraft, four of which lack a reward function. We clarified the benchmark, providing concrete evaluation recommendations and introducing another avenue for benchmarking: automated evaluations. Our dataset, BEDD, consists of two key parts: `Demonstrations Dataset` and `Evaluation Dataset`. `Demonstrations Dataset` contains almost 14,000 videos showing successful task completions of the four reward-free tasks. The `Evaluation Dataset` consists of over 3,000 human evaluations to support the development of automated evaluations of hard-to-specify tasks. To our knowledge, this dataset is the largest of its type: one that supports both training and evaluating LfHF agents on tasks with hard-to-specify reward functions.

We demonstrated the utility of our dataset and benchmark by presenting an analysis of the demonstrations and the several algorithms in the `Evaluation Dataset`. We showed that the algorithms exhibit varying degrees of performance on our tasks, as evaluated by human judges on a variety of factors. Our results suggest that there is ample room for improvement in learning from human feedback. With the inclusion of our accessible code for benchmarking and evaluating agents, we hope that our contributions will encourage the development of more effective approaches for both learning from human feedback and evaluating these techniques in the future.

## Acknowledgments and Disclosure of Funding

Creating these datasets and this benchmark was only possible with the help of many people and organizations. FTX Future Fund Regranting Program, Microsoft, Encultured AI, and AI Journal provided financial support for the competition and resulting data. We thank Berkeley Existential Risk Initiative (BERI) for the support in organizing the BASALT competition. Karolis Ramanauskas was supported by the UKRI Centre for Doctoral Training in Accountable, Responsible and Transparent AI (ART-AI) [EP/S023437/1] and the University of Bath. We thank our amazing advisory board of the 2022 competition — Fei Fang, Kianté Brantley, Andrew Critch, Sam Devlin, and Oriol Vinyals — for their advice and guidance. We also thank any previous organizers or advisors of the previous competitions for their contributions. Finally, we thank AIcrowd for their help hosting the competion and the MTurk workers for their work evaluating the submissions.

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
