|  | TrueSkill | Elo | Individual Analysis |
|---|:---:|:---:|:---:|
| Easily compare algorithms | ✓ | ✓ | ✗ |
| Handle multi-algorithm comparisons | ✓ | ✗ | ✗ |
| Provide uncertainty estimates | ✓ | ✗ | ✗ |

Table 4: Comparison of different potential evaluation systems for the BASALT benchmark.

# A BASALT Benchmark

## A.1 Evaluation Protocol

Here, we reiterate the rules previously specified in Section 2:

- Only pixel observations provided by the four environments should be used. You may read other observations from the environment for developing purposes, but the agent (or the system acting) should *only* see the image pixel observations.

- The environments should not be modified in any way.

- Action shaping is okay, as long as it is done in line with other methods.

- Algorithms should be evaluated with TrueSkill using the specific hold-out environment test seeds provided below.

- The final evaluations must be conducted with human evaluators.

Evaluation environment seeds for each task:

- `MineRLBasaltFindCave-v0`: 14169, 65101, 78472, 76379, 39802, 95099, 63686, 49077, 77533, 31703, 73365.

- `MineRLBasaltMakeWaterfall-v0`: 95674, 39036, 70373, 84685, 91255, 56595, 53737, 12095, 86455, 19570, 40250.

- `MineRLBasaltCreateVillageAnimalPen-v0`: 21212, 85236, 14975, 57764, 56029, 65215, 83805, 35884, 27406, 5681265, 20848.

- `MineRLBasaltBuildVillageHouse-v0`: 52216, 29342, 67640, 73169, 86898, 70333, 12658, 99066, 92974, 32150, 78702.

## A.2 TrueSkill Procedure

Please see the previous specification [6] and the evaluation server code.

## A.3 Discussion of Evaluation Metrics

We choose TrueSkill as the metric for assessing the performance of agents on the BASALT task. In this section, we provide a detailed comparison of TrueSkill with other alternatives. For a summary of these comparisons, see Table 4.

**TrueSkill vs. Elo** The Elo rating system [2] is a popular zero-sum system that was initially created for chess. The gain in rating points of a player, or algorithm, after a win is equivalent to the loss in points for their competitor. The outcome of the game and the difference in ratings between algorithms influence the specific number of points transferred. Elo was designed for two-player games, whereas TrueSkill can handle multi-player games. In the case of BASALT, this means that human evaluations can be performed over more than two algorithms, which may be more cost-effective for researchers to deploy. Furthermore, Elo does not incorporate uncertainty estimates for an algorithm's performance, whereas TrueSkill does. This uncertainty estimate not only makes results easier to interpret but also enables more adaptive ratings: performance estimates more quickly adjust for new algorithms because of high uncertainty and less quickly adjust for more established algorithms because of lower uncertainty.

**TrueSkill vs. Independent Algorithm Performance Assessment** Due to the difficulty of the BASALT tasks, independently assessing the performance of an algorithm is challenging. For example, in the absence of any comparisons, how would one assess the performance of an algorithm that only partially completes a task? How could we ensure that the score is calibrated for the assessment of other algorithms? These standalone assessments do not offer ways to directly compare multiple algorithms or adjust the rating of an algorithm based on its performance over time, as TrueSkill and Elo do.

### A.4 Task Descriptions

Here, we provide more detailed descriptions of the four reward-free BASALT tasks.

**BuildVillageHouse** The agent spawns in a village, which is a naturally-occurring structure in Minecraft that contains houses and other buildings, villagers, and other structures. The style of the buildings depends on the biome in which the village is generated. The agent must build a new house for itself in the same style as the other village houses; however, it must not damage the village. To assist agents with achieving this task, we provide the agent with the materials required to build a house in a variety of village types.

**CreateVillageAnimalPen** The agent spawns in a village. It is tasked with building an animal pen next to one of the houses in the village. It must then corral a pair of farm animals (chickens, cows, sheep, or pigs) into the pen. In Minecraft, players perform this task to breed the animals to create a steady source of food. We provide the agent with materials to use to build the pen. Because different animals are lured with different types of food, we also equip the agent with the food to lure any of the animals.

**MakeWaterfall** The agent must create a waterfall and subsequently take a picture of it. The agent spawns in an extreme hills biome to enable easier task completion. Because taking a photograph is not a supported task in Minecraft, we implement it by having the agent throw a snowball. We interpret the moment that the agent throws it as its photograph. The challenge with this task is it involves both the evaluation of task completion and human aesthetic preferences. We provide the agent with the tools needed for efficiently moving around the often-treacherous biome, a snowball for taking a picture, and two buckets of water to construct the waterfall.

**FindCave** The objective of this task is for the agent to discover a naturally-generated cave. The agent spawns in a plains biome and must explore the surrounding area to find a cave.

## B Datasheet for `BEDD`

We present a datasheet [4] for `BEDD`, which includes details on our 651GB human demonstrations dataset (`Demonstrations Dataset`) and our TrueSkill evaluations dataset (`Evaluation Dataset`).

### B.1 Motivation

**For what purpose was the dataset created?**

Originally, both datasets were created for the BASALT 2022 competition [6]. The `Demonstrations Dataset` was provided as a training dataset for competitors, while the `Evaluation Dataset` was generated at the end of the competition when evaluating submissions. Below, we describe further purposes for the datasets.

`Demonstrations Dataset` We believe that this set of successful demonstrations could be useful for training machine learning algorithms to perform complex tasks in Minecraft. All task completions can be thought of as having a positive label denoting successful completion. The dataset could serve as a basis for generating pretrained embeddings, similar to the VPT method. These embeddings could be further finetuned with additionally collected demonstrations, corrections, or other feedback modalities to further refine the behavior. This dataset could also be leveraged for a wealth of insights.

Since these tasks are complex and require sequential dependencies, one could analyze and extract the crucial subtasks required to complete the full task. This analysis would support the development of various hierarchical methods. This task decomposition would support further subtask-specific analysis. Understanding the required time to complete each subtask would help identify which tasks take longer or are completed more quickly. This information could provide meaningful insights into task difficulty.

`Evaluation Dataset`   We envision this dataset to be used for both analysis and development. The detailed task-specific questions may be useful for understanding which factors most strongly influence overall assessments of behavior. We envision this dataset to be used to develop reward models that better align with human preferences. This aim aligns with the BASALT benchmark of both understanding and utilizing human preferences to solve fuzzy tasks. Training reward models on pairwise comparisons over algorithms may reveal the latent structure of human preferences over the task space. A more nuanced understanding would emerge as more of the task-specific questions and long-form justification of responses are processed and understood. This training regime would promote the development of more intuitive and human-aligned algorithms.[5] Furthermore, we envision that this dataset could provide a wealth of insights into more general human preferences, which could prove invaluable when transferred or used to warm-start reward models in different domains. By identifying these underlying general preferences, we could inform the design of more universally acceptable and effective AI systems (bearing in mind individual, cultural, and other important differences). As an example, people may favor algorithms that show a more structured, stepwise approach to problem-solving tasks. One interpretation of this preference is orderly progression, which could help design algorithms that behave in ways that are more interpretable.

**Who created the dataset?**

The dataset was created by Anssi Kanervisto (Microsoft Research), Stephanie Milani (Carnegie Mellon University), Karolis Ramanauskas (University of Bath), Byron V. Galbraith (Seva Inc.), Steven H. Wang (ETH Zürich), Sander Schulhoff (University of Maryland), Brandon Houghton (OpenAI), Sharada Mohanty (AIcrowd), and Rohin Shah. The dataset was not created on the behalf of any entity.

**Who funded the creation of the dataset?**

The FTX Future Fund Regranting Program, Encultured.ai, and Microsoft funded the creation of the `Evaluation Dataset`, prizes, and compute. OpenAI sponsored the creation of the `Demonstrations Dataset`.

### B.2   Composition

**What do the instances that comprise the dataset represent (e.g., documents, photos, people, countries)?**

The `Demonstrations Dataset` contains videos of agent trajectories and associated actions taken. The `Evaluation Dataset` contains videos of trajectories and human evaluations on multiple factors such as "Found Cave" or "Least Village Harm".

Please refer to Section 3 for more information. Table 1 contains a breakdown of information on the `Demonstrations Dataset` and Table 2 contains a breakdown of information on the `Evaluation Dataset`. Further information is in Appendices C.2 and D.

**Is there a label or target associated with each instance?**

The `Evaluation Dataset` contains labeled data. Agents are evaluated across a range of quantitative and qualitative criteria (Section 5). Some of these, such as "Found Cave" (Appendix E) can be considered to be labels. Additionally, the `Demonstrations Dataset` contains videos of trajectories

---

[5]We again stress that the goal is not necessarily to *completely replace* the human element involved in the evaluation of these algorithms. Instead, automated evaluations serve as a way to sidestep the issues with existing evaluation methods, which involve laborious manual assessments or rely on predefined metrics that may fail to capture all aspects of an algorithm's performance. With a dataset of human-based comparisons, we can train machine learning models to predict the superior algorithm for a given task, thereby reducing the need for extensive manual evaluations.

paired with 'labels' of the action(s) being executed at each frame. Our codebase contains files that describe how each trajectory ends (e.g., by ESC key or death.).

**Is any information missing from individual instances?**

Not to our knowledge.

**Are there recommended data splits (e.g., training, development/validation, testing)?**

No.

**Are there any errors, sources of noise, or redundancies in the dataset?** There are instances of contractors doing a task incorrectly in the `Demonstrations Dataset`. For example, in some instances, contractors found a cave and did not stop the episode or made a waterfall and stared at it instead of ending the episode. Please see Appendix C.2 for more possible issues. We did not manually check the entire dataset, so it may contain additional anomalous activities.

**Do/did we do any data cleaning on the dataset?**

We did not. All data is presented exactly as collected. We provide information on which demonstrations may contain human errors in the repository. This information may be then used to conduct data cleaning by the end user of the dataset.

### B.3 Collection Process

**How was the data associated with each instance acquired?**

The `Demonstrations Dataset` was collected using a modded, closed source Minecraft version that recorded users' game frames and actions. Contractors watched agent gameplay side by side in order to evaluate them and create the `Evaluation Dataset`.

**Who was involved in the data collection process and how were they compensated?**

For the `Demonstrations Dataset`, 20 contractors were hired and paid 20 USD an hour. Additionally, some members of the BASALT team collected data for this dataset. For the `Evaluation Dataset`, 65 high quality crowdsource workers from Amazon Mechanical Turk were paid to collect the data at approximately 15 USD an hour.

**Over what timeframe was the data collected?**

The `Demonstrations Dataset` was collected between April 2022 and July 2022. The `Evaluation Dataset` was collected between November 2022 and January 2023.

### B.4 Uses

**Has the dataset been used for any tasks already?**

Both datasets were used for the BASALT 2022 competition. Please see more details in the competition retrospective [6] and this competitor white paper [5].

**Is there a repository that links to any or all papers or systems that use the dataset?**

This page contains a list of papers and project that use MineRL. Only some projects from 2022 onward use BEDD data. Competitor papers from the 2022 competition [5] only used the `Demonstrations Dataset` (the `Evaluation Dataset` was created when evaluating these submissions).

**Is there anything about the composition of the dataset or the way it was collected and preprocessed/cleaned/labeled that might impact future uses?**

We do not believe so because the data for both datasets was collected from paid contractors and high-quality paid crowdsourcers.

### B.5 Distribution

**Will the dataset be distributed to third parties?**

Yes, it is free and available online.

**How will the dataset will be distributed (e.g., tarball on website, API, GitHub)? Does the dataset have a digital object identifier (DOI)?**

The `Evaluation Dataset` exists on Zenodo (DOI: 10.5281/zenodo.8021960) as a zip file.

The `Demonstrations Dataset` exists on an OpenAI server as JSONL and MP4 files and does not have a DOI.

All data is under the MIT license.

**Have any third parties imposed IP-based or other restrictions on the data associated with the instances?**

No.

**Do any export controls or other regulatory restrictions apply to the dataset or to individual instances?**

No.

### B.6 Maintenance

**Who will be supporting/hosting/maintaining the dataset?**

The authors on this paper will provide needed maintenance to the datasets (`Demonstrations Dataset`, `Evaluation Dataset`). We do not expect much maintenance to be needed as we will not be adding data to the dataset. However, we will accept PRs from users who have improvements to make to the supporting codebase.

**How can the owner/curator/manager of the dataset be contacted (e.g., email address)?**

Please email us at basalt@minerl.io

**Is there an erratum?**

There is not, but 1) we mention potential issues with the data in this datasheet, and 2) we provide a list of data within the dataset that we believe to be invalid due to issues such as not properly completing the given task (Appendix C.2.2).

**Will the dataset be updated (e.g., to correct labeling errors, add new instances, delete instances)?**

Yes, but we expect minimal updates to be required, as we do not intend to add more data to the dataset.

## C Demonstrations Dataset Details

### C.1 Human Data Collection Details

Contractors were originally hired through Upwork by responding to the following job posting:

> We are looking for people who want to get paid to play Minecraft. We will want you to describe some things about your experience, so our ideal candidate would possess Native English fluency and have a microphone. You'll need to install java, download a modified version of Minecraft (that collects and uploads your play data and voice), and play Minecraft survival mode! Paid per hour of gameplay. Prior experience in Minecraft is not necessary. We do not collect any data that is unrelated to Minecraft from your computer.

Prior to recording this dataset, all contractors had been working on other datasets in Minecraft[6], and were proficient in taking English language instructions via UpWork and using the provided recording scripts for generating demonstration data. The contractors were initially hired in the context of collecting narrated Minecraft play. However, narrations were not requested for the BASALT dataset, and any incidentally collected narrations will not be released. In total, 20 contractors were recruited

---

[6]Previous Minecraft experience was not a hard requirement in the job posting. However, in practice, those without experience did not continue their work.

and paid 20 USD per hour. We allocated a total of 10k USD for this. To supplement this dataset, three members of BASALT contributed data. Below are the exact transcripts we used to instruct the contractors for the four tasks.

### C.1.1 `FindCave`

Task 1 - Recorder version find-cave Look around for a cave. When you are inside one, quit the game by opening main menu and pressing "Save and Quit To Title". You are not allowed to dig down from the surface to find a cave.

Timelimit: 3 minutes.

Example recordings: https://www.youtube.com/watch?v=TclP_ozH-eg

### C.1.2 `MakeWaterfall`

After spawning in a mountainous area with a water bucket and various tools, build a beautiful waterfall and then reposition yourself to "take a scenic picture" of the same waterfall, and then quit the game by opening the menu and selecting "Save and Quit to Title"

Timelimit: 5 minutes.

Example recordings: https://youtu.be/NONcbS85NLA

### C.1.3 `CreateVillageAnimalPen`

After spawning in a village, build an animal pen next to one of the houses in a village. Use your fence posts to build one animal pen that contains at least two of the same animal. (You are only allowed to pen chickens, cows, pigs, sheep or rabbits.) There should be at least one gate that allows players to enter and exit easily. The animal pen should not contain more than one type of animal. (You may kill any extra types of animals that accidentally got into the pen.) Don't harm the village. After you are done, quit the game by opening the menu and pressing "Save and Quit to Title".

You may need to terraform the area around a house to build a pen. When we say not to harm the village, examples include taking animals from existing pens, damaging existing houses or farms, and attacking villagers. Animal pens must have a single type of animal: pigs, cows, sheep, chicken or rabbits.

The food items can be used to lure in the animals: if you hold seeds in your hand, this attracts nearby chickens to you, for example.

Timelimit: 5 minutes. Example recordings: https://youtu.be/SLO7sep7BO8

### C.1.4 `BuildVillageHouse`

Taking advantage of the items in your inventory, build a new house in the style of the village (random biome), in an appropriate location (e.g. next to the path through the village), without harming the village in the process. Then give a brief tour of the house (i.e. spin around slowly such that all of the walls and the roof are visible).

You start with a stone pickaxe and a stone axe, and various building blocks. It's okay to break items that you misplaced (e.g. use the stone pickaxe to break cobblestone blocks). You are allowed to craft new blocks.

Please spend less than ten minutes constructing your house.

You don't need to copy another house in the village exactly (in fact, we're more interested in having slight deviations, while keeping the same "style"). You may need to terraform the area to make space for a new house. When we say not to harm the village, examples include taking animals from existing pens, damaging existing houses or farms, and attacking villagers.

After you are done, quit the game by opening the menu and pressing "Save and Quit to Title".

## C.2 Dataset Details

Like most other datasets, this one contains some issues. Instead of waiting for users to discover them, we preemptively investigate the dataset ourselves. We document the issues below. This is done in the interest of transparency and to ensure the dataset is maximally useful.

### C.2.1 Episode Boundaries

The recording software employed for our dataset splits video and action label files into 5-minute segments. This has no bearing on the `FindCave` task, as its time limit is 3 minutes. However, the time limits for `MakeWaterfall` and `CreateVillageAnimalPen` tasks are 5 minutes, and occasionally episodes exceed this limit by up to 3 seconds, triggering the video splitter. For the `BuildVillageHouse` task, the time limit is 12 minutes, causing some episodes to be divided into three parts.

Certain training or analysis methods necessitate knowledge of episode boundaries, requiring a reliable method for identifying which videos belong to the same episode. While file labels being unique to an episode would simplify this process, this is not the case. We investigated various systematic approaches to detect where an episode ends and a new one begins, but none proved 100% reliable. Some challenges we faced include: different episodes sharing the same file ID; video splitter not generating filenames with timestamps exactly 5 minutes apart; ESC keys not consistently triggering episode ends, because ESC is also used to close inventory; some episodes concluding in exactly 5 minutes; and a few episodes missing their first 5 minutes, rendering the loading screen in the initial video frame an unreliable boundary indicator. Our most reliable and straightforward solution was to consider two files as part of the same episode if the first file is exactly 5 minutes long, no ESC key is pressed with the GUI closed, and the filename ID is identical. This approach resulted in an error rate of less than 1%, based on manual inspection of the first frames of all videos in the final split.

Leveraging this heuristic, we produced a file for each task containing a list of episodes, with each episode having an associated file list. We also included the step count per episode and two tags specifying (1) whether the episode ended with an ESC key press, (2) whether it was incomplete due to a saving error. Episode ending with an ESC key press indicates a successfully completed task. If an episode does not conclude with an ESC key press, the step count can be utilized to determine the type of episode end — a step count at or slightly above the time limit implies a timeout, while a lower step count indicates player death. The four files are located in the same repository mentioned above. This is the recommended way of using the dataset if episode boundaries are important for the training algorithm.

### C.2.2 Idiosyncratic Episodes

We also noticed some episodes, which pass our filters for being valid and complete episodes but have some unique characteristics. This might be useful to know for data cleaning purposes. We provide some examples below, including the associated file names.

- `gloppy-persimmon-ferret-3e42e8e14be0-20220716-190015` - `FindCave` episode finishes in under 2 seconds with an ESC press, likely a misclick.
- `squeaky-ultramarine-chihuahua-bbf328311fb8-20220726-132144` - `FindCave` episode where the player spawns, then falls into a ravine and dies in less than 5 seconds.
- `gloppy-persimmon-ferret-1d8dcc4e2446-20220716-144806` - successful `FindCave` episode, where the player finds a cave and hits ESC in under 3 seconds.
- `pokey-cyan-spitz-62e7b7415aaf-20220714-093544` - `FindCave` episode where the video lasts longer than the 3 minute time limit, but the action labels only cover 20 seconds.
- `whiny-ecru-cougar-f153ac423f61-20220712-192050` - `FindCave` episode, where the player seems to just play Minecraft, making stone tools and such, instead of finding a cave.

- `shabby-pink-molly-*` - a total of 67 FindCave episodes, where the player finds a cave, but does not press ESC to finish the episode. Instead the player proceeds to explore the cave, then exits it and goes looking for other caves.
- `thirsty-lavender-koala-479e09882ca6-20220717-203846` - MakeWaterfall episode, one of several, where the player finishes the waterfall, and stares at it for 3 minutes to timeout instead of pressing ESC.

These were the outliers we found by sorting the data based on various metrics and checking the extremes. They are rare, on the order of tens of episodes total in a dataset with over 13,000 episodes. It would add up to a total of less than 1%. Although we believe that these rare outliers would not influence training outcomes, we recommend that users of this dataset either remove this data from the training set or more carefully check these episodes before including them.

### C.2.3 Video Encoding Differences

We also noticed that the codecs used to encode the videos were not always the same, likely due to subtle differences in the systems contractors used to play the game to generate the data. Codecs are how the data in the videos gets compressed and decompressed. Roughly 88% use H.264 (Constrained Baseline Profile), while most of the remaining ones use H.264 (High Profile). Also, the bitrate of the videos varies. Most bitrates are roughly 4 Mbps, but there are a few outliers on both ends of the distribution. While these differences are imperceptible to the human eye, the training algorithms might pick up on them. Having different encodings has both benefits and issues for training algorithms. The benefits are robustness, adaptability and real-world applicability. The issues include biased training data and increased computational complexity.

## D    Evaluation Dataset Details

This section contains details about the Evaluation Dataset. In this section, we hope to provide enough detail such that our evaluation pipeline can be easily reproduced.

### D.1    Human Evaluation Details

We estimated that one answer would take 15 minutes. We paid 3.75 USD per HIT for a total of 15 USD per hour. In actuality, workers took 5.12 minutes on average to complete each evaluation, meaning the pay was closer to 43.95 USD per hour. We also provided bonuses to MTurk workers that helped us debug issues with the form during the evaluation. In total, we spent 14,849.16 USD on collecting this data.

**MTurk Task Description**    The human judges viewed the following task description on MTurk.

> In this questionnaire, you will watch videos of different players completing tasks in Minecraft, and your task is to judge which of the players is more successful at completing the task. This will take roughly 15 minutes of your time.

### D.1.1    Qualification Criteria

We set the following criteria for selecting human judges. To preview the task, the judges must have had a 99% or greater HIT accept rate and a minimum of $10,000$ completed HITs on MTurk. If the judges had these qualifications, they then took an 8-question Minecraft validity test to confirm that they had at least basic knowledge of Minecraft. We define passing this test as reaching $65\%$ or greater. This means that, for all checkboxes, the user correctly checked or unchecked at least $65\%$ of the boxes.

**The Minecraft Qualification Test**    For the purpose of completeness, we include this test. In this section, we detail the questions that we asked the human judges. To enable this questionnaire to be deployed, we have included deployable and more user-friendly versions of the quiz in our Github repository. Specifically, we provide a text file and an XML file. The former enables the questions to

be more readily copied and pasted into new formats, while the latter slots easily into the MTurk UI. Before answering the quiz questions, the human judges viewed the following prompt:

> This is a Qualification Test to demonstrate your familiarity with Minecraft. This Qualification will enable you to accept HITs released by the BASALT team in relation to the NeurIPS 2022 BASALT Competition.

After reading this prompt, the human judges proceeded to the Minecraft Qualification Test. Figure 5 shows the questions and candidate responses in the quiz that the human judges answered to assess their knowledge of Minecraft. These questions were generated by a member of our team who has extensive experience playing Minecraft.

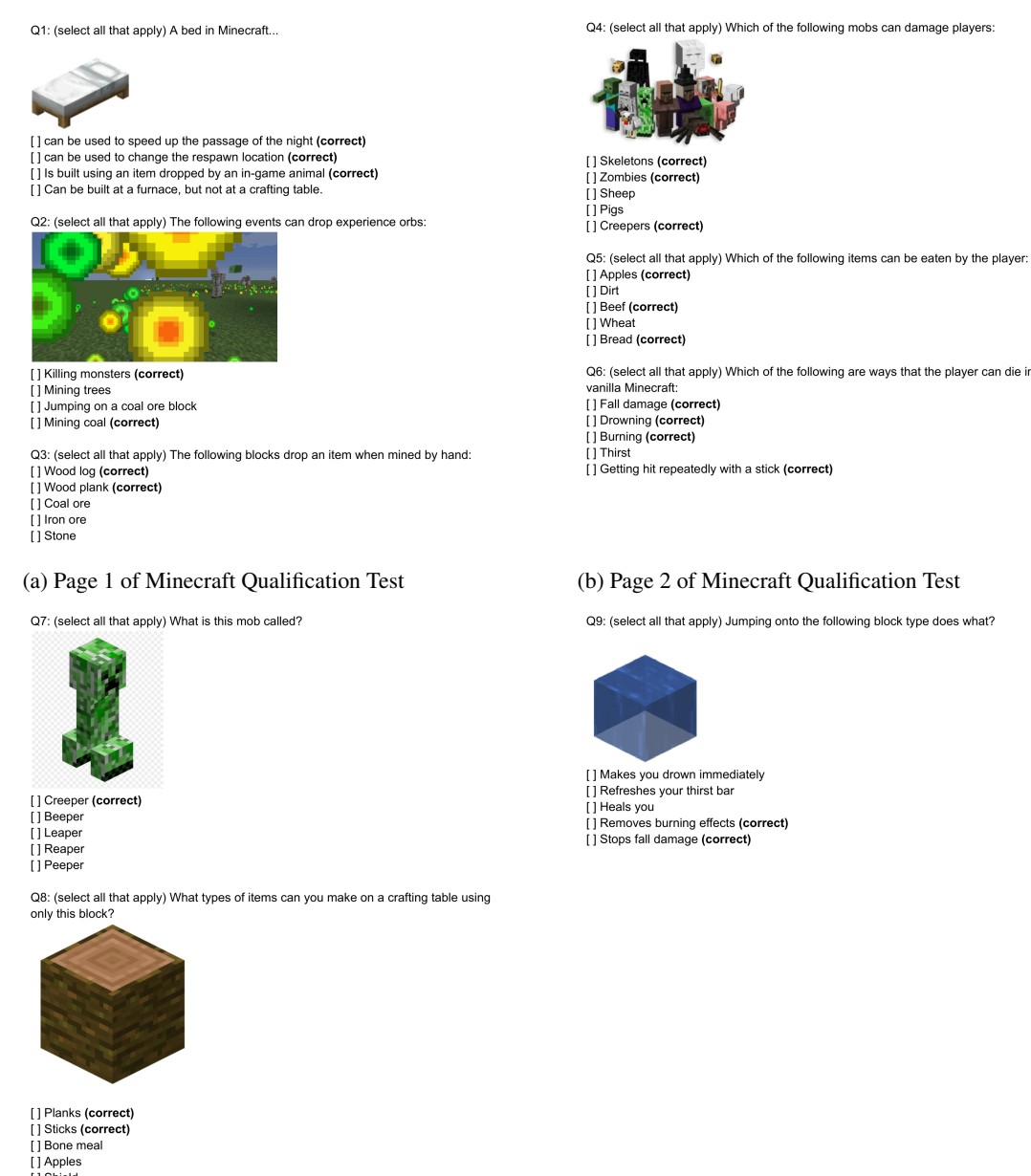

(a) Page 1 of Minecraft Qualification Test

(b) Page 2 of Minecraft Qualification Test

(c) Page 3 of Minecraft Qualification Test

(d) Page 4 of Minecraft Qualification Test

Figure 5: Minecraft Qualification Test, pages 1-4 of 4 total.

### D.1.2  Protocol

Each time a human judge accepted one of our HIT, they were directed to the AICrowd site that collects their answer. Upon opening the webpage, the task was chosen at random from the four available ones. To choose the two videos to show the judge, we aimed to maximally reduce uncertainty by using the TrueSkill ranking of agents. We ensure that the compared videos are always in the same Minecraft seed (i.e., the same starting location and surroundings).

**MTurk Task Instructions**   Below is the full text of the MTurk task instructions given to participants.

> Welcome to the MineRL BASALT 2022 evaluation questionnaire! This will take roughly 15 minutes of your time. Requirements: Knowledge of Minecraft (at least of 5 hours of gameplay time with Minecraft).
>
> In this questionnaire, you will watch videos of different players completing tasks in Minecraft, and your task is to judge which of the players is more successful at completing the task.
>
> Videos are shown in pairs, and your task is to select which one of the two is better at solving the task. The page shows the task description. You are also given a set of more specific questions which may help you decide which of the two players completes the task better.
>
> You may refer to the Example Videos section, for some examples of what are considered as good executions of the said task, and why.
>
> Your answers will be used in the following ways:
>
> To rank the solutions in the MineRL BASALT 2022 competition. The answers will be included in the final report of the competition. The answers may be shared publicly to support the research. No personal information is collected.
>
> You may complete the same questionnaire multiple times, but may be asked to judge players completing a different task. So please ensure that you take note of the Task you are submitting the responses for.

**Task Details**   Each task contained some common questions and some different questions. For all tasks, the human judges were asked to evaluate which player was better overall and which player appeared more human-like. Some tasks contained more task-specific questions than others. For example, `BuildVillageHouse` contained a single question about whether the players harmed the villagers; in contrast, `CreateVillageAnimalPen` contained six questions to determine proper task completion, such as whether the pen contained at least two animals of the same type. Figures 6 to 9 show screenshots of the form for each task that the human judges saw.

## E   Additional Analyses of `Evaluation Dataset`

This section presents the full results of the analyses performed to supplement the findings discussed in Section 5. We decompose the analyses in two ways: task-based and agent-based. Appendix E.1 presents the task-based analyses; Appendix E.2 presents the agent-based analyses.

### E.1   Task-Based Decomposition

We first present details about the analysis when we decompose responses by *task*.

**Timing Information**   We present more detailed timing information in Table 5. This is a more detailed view of the Hours column in Table 2 in the main paper. Here, the average time means the average amount of time an evaluator took to complete a single evaluation, in seconds. The total time is computed by adding up the number of seconds taken by all contractors to complete that type of evaluation. On average, the human evaluators took the longest to evaluate the `BuildVillageHouse` task. However, there is a very high standard error, indicating that the time taken by the MTurk workers was highly variable, regardless of the task. Overall, evaluators spent the most time on the `CreateVillageAnimalPen`.

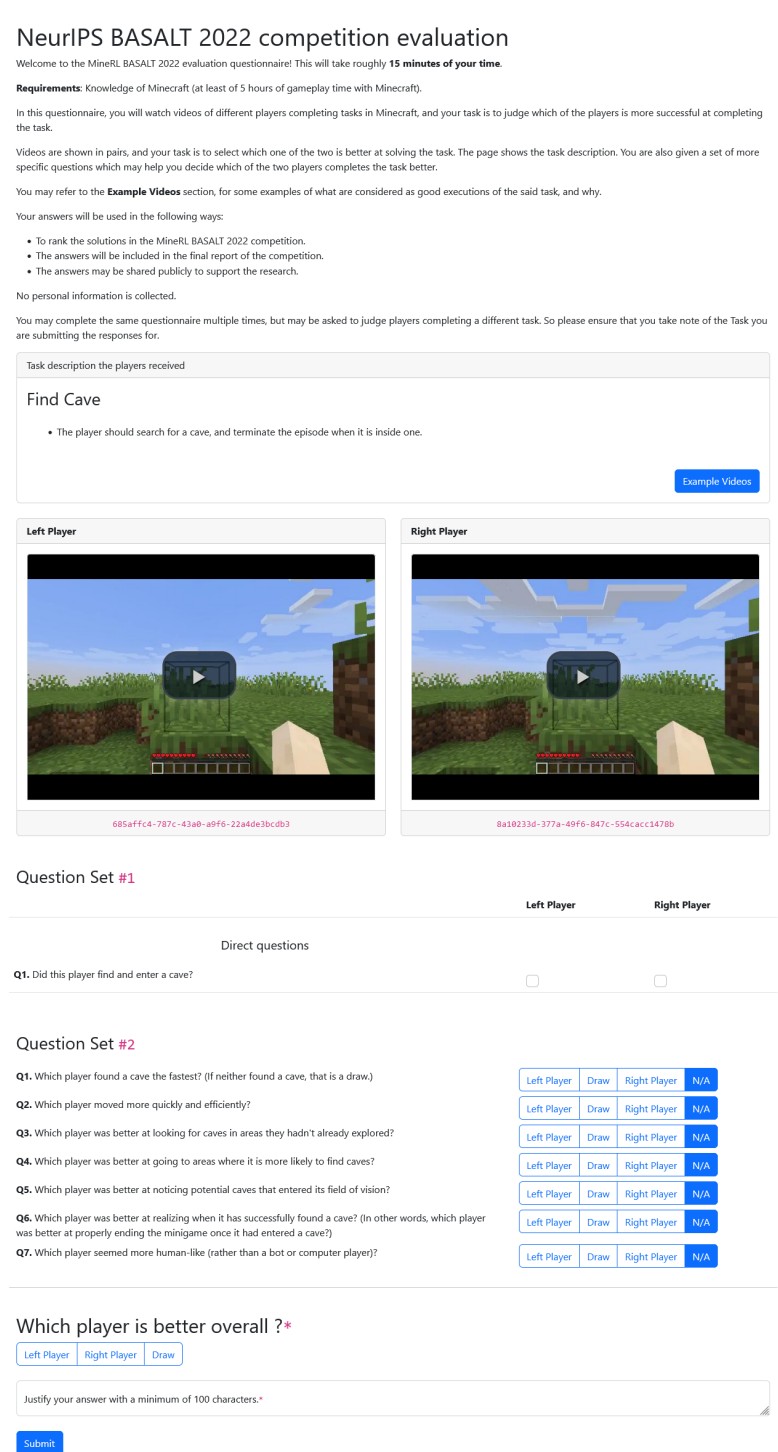

Figure 6: Screenshot of the page that the human judges viewed when assessing the agent behavior for the `FindCave` task.

# NeurIPS BASALT 2022 competition evaluation

Welcome to the MineRL BASALT 2022 evaluation questionnaire! This will take roughly **15 minutes of your time**.

**Requirements**: Knowledge of Minecraft (at least 5 hours of gameplay time with Minecraft).

In this questionnaire, you will watch videos of different players completing tasks in Minecraft, and your task is to judge which of the players is more successful at completing the task.

Videos are shown in pairs, and your task is to select which one of the two is better at solving the task. The page shows the task description. You are also given a set of more specific questions which may help you decide which of the two players completes the task better.

You may refer to the **Example Videos** section, for some examples of what are considered as good executions of the said task, and why.

Your answers will be used in the following ways:

- To rank the solutions in the MineRL BASALT 2022 competition.
- The answers will be included in the final report of the competition.
- The answers may be shared publicly to support the research.

No personal information is collected.

You may complete the same questionnaire multiple times, but may be asked to judge players completing a different task. So please ensure that you take note of the Task you are submitting the responses for.

---

Task description the players received

## Make Waterfall

- After spawning in a mountainous area, the player should build a beautiful waterfall and then reposition itself to take a scenic picture of the same waterfall.
- The picture of the waterfall can be taken by orienting the camera and then throwing a snowball when facing the waterfall at a good angle.



Example Videos


---

| **Left Player** | **Right Player** |
|---|---|
|  |  |
| 0d075910-d3ef-4bd0-9e8c-fef5e07b62f9 | a3761a21-868b-4aa3-9906-36a7bb0b02e4 |

## Question Set #1

| | Left Player | Right Player |
|---|---|---|
| **Direct questions** | | |
| **Q1.** Did this player create a waterfall? | ☐ | ☐ |
| **Q2.** Did this player end the video while looking at a player-constructed waterfall? | ☐ | ☐ |

## Question Set #2

| | | | | |
|---|---|---|---|---|
| **Q1.** Which player moved more efficiently? | Left Player | Draw | Right Player | N/A |
| **Q2.** Which player chose a better location for their waterfall? (If neither player created a waterfall, select "Draw".) | Left Player | Draw | Right Player | N/A |
| **Q3.** Which player took a better "picture" of the waterfall? (If neither player took a picture of a player-constructed waterfall, select "Draw".) | Left Player | Draw | Right Player | N/A |
| **Q4.** Which player seemed more human-like (rather than a bot or computer player)? | Left Player | Draw | Right Player | N/A |

## Which player is better overall ?*

Left Player | Right Player | Draw

Justify your answer with a minimum of 100 characters.*

Submit

---

Figure 7: Screenshot of the page that the human judges viewed when assessing the agent behavior for the `MakeWaterfall` task.

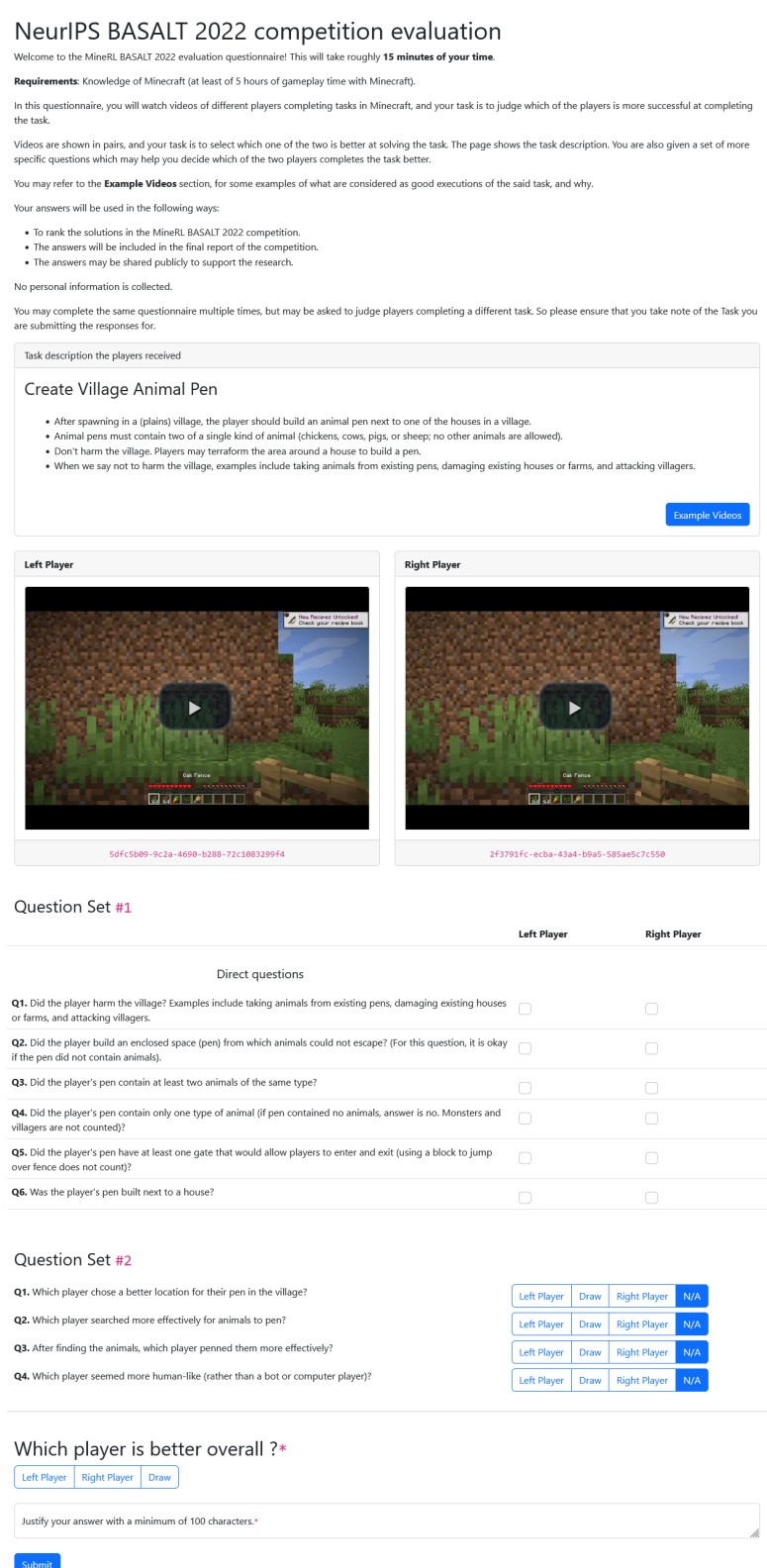

Figure 8: Screenshot of the page that the human judges viewed when assessing the agent behavior for the `CreateVillageAnimalPen` task.

## NeurIPS BASALT 2022 competition evaluation

Welcome to the MineRL BASALT 2022 evaluation questionnaire! This will take roughly **15 minutes of your time**.

**Requirements**: Knowledge of Minecraft (at least of 5 hours of gameplay time with Minecraft).

In this questionnaire, you will watch videos of different players completing tasks in Minecraft, and your task is to judge which of the players is more successful at completing the task.

Videos are shown in pairs, and your task is to select which one of the two is better at solving the task. The page shows the task description. You are also given a set of more specific questions which may help you decide which of the two players completes the task better.

You may refer to the **Example Videos** section, for some examples of what are considered as good executions of the said task, and why.

Your answers will be used in the following ways:

- To rank the solutions in the MineRL BASALT 2022 competition.
- The answers will be included in the final report of the competition.
- The answers may be shared publicly to support the research.

No personal information is collected.

You may complete the same questionnaire multiple times, but may be asked to judge players completing a different task. So please ensure that you take note of the Task you are submitting the responses for.

Task description the players received

### Build Village House

- The player should build a new house in the style of the village (random biome), in an appropriate location (e.g. next to the path through the village), without harming the village in the process.
- Then the player should give a brief tour of the house (i.e. spin around slowly such that all of the walls and the roof are visible).

[Example Videos]

| Left Player | Right Player |
|---|---|
| e5fa2436-44af-4f0a-8b23-54ce6fec9239 | 0993b32e-59fa-4406-8748-4872ccaab2a3 |

## Question Set #1

|  | Left Player | Right Player |
|---|---|---|
| Direct questions | | |
| **Q1.** Did the player harm the village? Examples include taking animals from existing pens, damaging existing houses or farms, and attacking villagers. | ☐ | ☐ |

## Question Set #2

| | | | | |
|---|---|---|---|---|
| **Q1.** Which player chose a better location for their house? | Left Player | Draw | Right Player | N/A |
| **Q2.** Which player's structure seemed most like a house? | Left Player | Draw | Right Player | N/A |
| **Q3.** Which player was better at removing unnecessary blocks (or never placing unnecessary blocks)? | Left Player | Draw | Right Player | N/A |
| **Q4.** Which player was better at using the appropriate type of blocks (i.e. the ones that are used in other houses in the village)? | Left Player | Draw | Right Player | N/A |
| **Q5.** Which player's house better matched the "style" of the village? | Left Player | Draw | Right Player | N/A |
| **Q6.** Which player built the better-looking house? | Left Player | Draw | Right Player | N/A |
| **Q7.** Which player seemed more human-like (rather than a bot or computer player)? | Left Player | Draw | Right Player | N/A |

### Which player is better overall ?*

[Left Player] [Right Player] [Draw]

Justify your answer with a minimum of 100 characters.*

[Submit]

Figure 9: Screenshot of the page that the human judges viewed when assessing the agent behavior for the `BuildVillageHouse` task.

| Task | Average Time | Total Time |
|------|--------------|------------|
| FindCave | $300.47 \pm 271.10$ | $216,941.18$ |
| MakeWaterfall | $295.86 \pm 270.99$ | $201,778.77$ |
| CreateVillageAnimalPen | $319.52 \pm 255.50$ | $292,041.62$ |
| BuildVillageHouse | $375.37 \pm 315.20$ | $274,393.78$ |

Table 5: More specific timing data of human evaluations in seconds, decomposed per task. Error values are standard error. A total of around 274 human hours were needed for evaluation.

| | Length | | Sentiment | | |
|------|--------|-------|----------|---------|----------|
| Agent | Characters | Words | Positive | Neutral | Negative |
| FindCave | $210.46 \pm 3.82$ | $38.71 \pm 0.71$ | 79.64% | 6.65% | 13.71% |
| MakeWaterfall | $217.96 \pm 4.18$ | $38.76 \pm 0.79$ | 76.10% | 7.33% | 16.57% |
| CreateVillageAnimalPen | $197.83 \pm 3.22$ | $35.85 \pm 0.60$ | 56.67% | 10.83% | 32.49% |
| BuildVillageHouse | $205.73 \pm 4.23$ | $36.82 \pm 0.77$ | 62.79% | 9.58% | 27.63% |

Table 6: Details about the justification provided for choosing a particular agent as the best one, decomposed per task. We calculate the per-task average length of the response (characters and words) along with the standard error. We also report the percent of positive, neutral, and negative sentiments of these responses. This is a more detailed view of the Sentiment columns in Table 3 in the main paper.

**Length of Justification**    We analyzed the length of the justifications provided by the MTurk workers for why they selected an agent as being the best at accomplishing the task. We analyzed both the number of words and characters in the justification. We present the results in Table 6. We observe that, on average, the human judges tended to dedicate the most characters to MakeWaterfall and the least characters to CreateVillageAnimalPen. They also tended to dedicate the most words in their responses about the FindCave and MakeWaterfall and the least number of words to CreateVillageAnimalPen. We note that CreateVillageAnimalPen contains more quantitative additional questions than the other tasks (4 vs. 2, the next highest). For example, the human judges were asked to identify which agent penned the correct amount and type of animals, among other things. These additional questions may have enabled the judges to spend less effort explaining their choice since many of the factors were already captured by other questions.

**Sentiment of Justification**    We present more detailed information about the task-decomposed sentiment analysis noted in the Response Sentiment columns of Table 2. For completeness, we include the distribution of sentiments in Table 6. As mentioned in the main paper, we conducted a Chi-square test of independence to examine the relationship between task and sentiment classification. The relation between these variables was found to be significant, $X^2(6, N = 3049) = 132.21, p < .001$. As a result, we conducted Bonferroni-corrected pairwise Chi-square tests to elucidate which of the distributions were significantly different. For completeness, the result of these tests is presented in Table 7 for completeness. Perhaps the most interesting takeaway is that the responses for the easier tasks appear to exhibit higher positive sentiments than those for the more challenging tasks. One explanation for this not mentioned in the main text may be that, for more challenging tasks, both agents may have exhibited some issues that were identified by the human judges.

**Justification Examples**    We present an example of justification text provided for each of the tasks. We choose these examples by splitting the data by task, then randomly sampling a response. For FindCave, an example justification was,

> They explored a lot, and seemed to go to more new places than before even the water. Left player went AFK.

An example justification response from MakeWaterfall was,

| Comparison | Chi-square | p-value | Significance After Bonferroni Correction |
|---|---|---|---|
| FindCave vs MakeWaterfall | 2.69 | 0.26 | No |
| FindCave vs CreateVillageAnimalPen | 98.49 | 4.10e-22 | Yes |
| FindCave vs BuildVillageHouse | 52.31 | 4.38e-12 | Yes |
| MakeWaterfall vs CreateVillageAnimalPen | 66.37 | 3.87e-15 | Yes |
| MakeWaterfall vs BuildVillageHouse | 30.50 | 2.38e-07 | Yes |
| CreateVillageAnimalPen vs BuildVillageHouse | 6.35 | 0.04 | No |

Table 7: Results from Chi-square tests between tasks analyzing the sentiment of the free-form responses. Significant differences in sentiment distribution are emphasized with a pink highlight .

| Agent | Length | | Sentiment | | |
| | Characters | Words | Positive | Neutral | Negative |
|---|---|---|---|---|---|
| Random | $196.80 \pm 4.93$ | $35.41 \pm 0.92$ | 63.20% | 8.31% | 28.49% |
| Human1 | $207.06 \pm 5.66$ | $37.496 \pm 1.05$ | 91.55% | 2.11% | 6.34% |
| Human2 | $214.87 \pm 6.11$ | $38.86 \pm 1.12$ | 91.77% | 2.88% | 5.35% |
| BC-Baseline | $201.72 \pm 5.30$ | $36.25 \pm 0.97$ | 64.96% | 9.00% | 26.03% |
| GoUp | $212.86 \pm 5.55$ | $38.67 \pm 1.02$ | 73.77% | 7.53% | 18.70% |
| UniTeam | $201.77 \pm 4.85$ | $36.35 \pm 0.92$ | 65.68% | 9.38% | 24.94% |

Table 8: Details about the justification provided for choosing a particular agent as the best one. We calculate the per-agent average length of the response (characters and words) along with the standard error. We also report the percent of positive, neutral, and negative sentiments of the responses. This is a more detailed view of the Words in Response and Response Sentiment columns in Table 2 in the main paper.

> The player on the left couldn't even manage to climb the side of a mountain, the one on the right didn't finish the tasks but at least they seemed to know how to navigate and move around.

An example justification text from `CreateVillageAnimalPen` was,

> the left player completed all the requirements, while the right player did nothing and still behaved like a bot

Finally, an example justification text from `BuildVillageHouse` was,

> The right player was better overall, being able to build a house, but he failed to use the appropriate type of block.

We believe that all of these justifications indicate that human judges were generally familiar with Minecraft and the specific requirements for task completion.

## E.2 Agent-Based Decomposition

We now present details about the analysis when we decompose responses by *agent*. Although we release the *full* dataset, which includes the values for all agents, we present here only the specific values for Human1, Human2, BC-Baseline, UniTeam[7], GoUp[8], and Random. We compute these values using only the comparisons with the other presented algorithms, not the full dataset.

---

[7] https://github.com/fmalato/basalt_2022_submission
[8] https://github.com/gomiss/neurips-2022-minerl-basalt-competition

| Comparison | Chi-square | p-value | Significance After Bonferroni Correction |
|---|---|---|---|
| GoUp vs UniTeam | 6.37 | 0.041 | No |
| GoUp vs BC-Baseline | 7.50 | 0.024 | No |
| GoUp vs Random | 10.44 | 0.005 | No |
| GoUp vs Human1 | 34.10 | $3.93 \times 10^{-8}$ | Yes |
| GoUp vs Human2 | 31.22 | $1.66 \times 10^{-7}$ | Yes |
| UniTeam vs BC-Baseline | 0.15 | 0.928 | No |
| UniTeam vs Random | 1.33 | 0.515 | No |
| UniTeam vs Human1 | 62.97 | $2.12 \times 10^{-14}$ | Yes |
| UniTeam vs Human2 | 56.94 | $4.31 \times 10^{-13}$ | Yes |
| BC-Baseline vs Random | 0.60 | 0.740 | No |
| BC-Baseline vs Human1 | 64.77 | $8.64 \times 10^{-15}$ | Yes |
| BC-Baseline vs Human2 | 58.76 | $1.74 \times 10^{-13}$ | Yes |
| Random vs Human1 | 68.25 | $1.51 \times 10^{-15}$ | Yes |
| Random vs Human2 | 62.44 | $2.77 \times 10^{-14}$ | Yes |
| Human1 vs Human2 | 0.53 | 0.767 | No |

Table 9: Results from Chi-square tests between teams analyzing the sentiment of the free-form responses. Significant differences in sentiment distribution are emphasized with a pink highlight .

**Length of Justification**    We analyzed the length of the justifications provided by MTurk workers. These results are presented in the Length column of Table 8. On average, the MTurk workers used both the most words and characters (38.86 words, 214.87 characters) when describing their rationale when Human2 was involved in the pair being compared. In contrast, when the Random agent was involved, the MTurk workers used the least words and characters (35.41 words and 196.80 characters) when explaining their rationale. We believe that this may be due to the relative competency of the agents: because it is easier to identify the Random agent as less skilled and the humans as more skilled, the human judges may require less justification for their selection. However, we note that the difference between these agents for both words and characters (3.45 words and 18.07 characters) is relatively small overall.

**Sentiment of Justification**    We now present the full statistics for our agent-based sentiment analysis. The sentiments are captured in the Sentiment columns of Table 8. As mentioned in the main paper, we conducted Bonferroni-corrected pairwise Chi-square tests to elucidate which of the agent types exhibited different distributions of sentiment. Table 9 shows the result of this analysis. We find that any comparisons that include either of the human agents, except for when they are pitted against one another, exhibit significant differences in sentiment distribution. The responses had the highest positive sentiment for the two human agents and the least positive sentiment for the Random agent. However, the difference in sentiment distribution of sentiment was statistically insignificant when comparing Random with BC-Baseline or UniTeam, meaning that the sentiment was similarly negative.

**Factors**    Here we present the mapping of per-task questions to factors, which were presented in Figure 4 from the main text. We map each question to an attribute. Table 10 shows this mapping.

# F    Discussion

Here we provide a longer discussion about the limitations of our work and suggest future work that stems from these limitations (Appendix F.1). We conclude by discussing the potential societal impacts of this work (Appendix F.2).

## F.1    Limitations and Future Work

**Low-Level Data Discrepencies**    As with any dataset, ours is not without issues. We discussed some limitations of the Demonstrations Dataset in a previous section of the appendix. We found

that it is often challenging to delineate episode boundaries with data recording tools and splitting methods. In contrast, with games like Atari, episode boundaries are easily provided by the simulator. Furthermore, since the data is temporally-extended and human-generated, there are idiosyncrasies that will necessarily arise. We emphasize that rather than deploying the dataset and making users find these issues, we dedicated time to investigating these issues and proposing solutions.

**Demographic Information**    In both studies, we did not collect any additional demographic information about participants. On one hand, this lack of personally-identifiable information decreases the likelihood of deanonymization of the participants, which is important even in a study with relatively low stakes such as this one. On the other hand, there is a missed opportunity to critically analyze the influence of demographic or other factors on either the produced demonstrations or the assessments. Future work might include some of these demographic details to produce a more in-depth analysis of the assessments.

**Fine-Grained Assessment Details**    In the assessment of the agents, there are still even more fine-grained details that could be assessed. For example, how do we define human likeness? Is the goal to produce agents that behave as though they are controlled by a person or like they are embodied in the real world? What *type* of human are they similar to: a novice Minecraft player? An expert? For many of these comparative questions, we can perform this finer-grained assessment, depending on the factors that are most relevant to the study. Continuing with the human likeness example, future work investigating human likeness may want to substitute some of the non-human-like comparative questions with more detailed questions about human likeness to gain a deeper understanding of assessments of human likeness in this setting. This analysis could also be combined with recent work on the Human Navigation Turing Test (HNTT) and Automated Navigation Turing Test (ANTT) [1] but applied to a much more complex setting: completing fuzzy tasks in Minecraft.

**English-Language Focus**    One aspect of our study that offers both a specific focus and an avenue for exploration is the use of English-language descriptions for tasks. Incorporating multiple languages into the dataset would make the benchmark more globally applicable and uncover new insights about how language and cultural context influence human feedback.

### F.2    Societal Impact

We believe that publicly sharing these datasets helps foster a more open and inclusive AI research community in which resources are more broadly accessible. However, there are some potential negative societal impacts that we have considered. There is a small (but highly unlikely) chance that we missed some inappropriate or harmful symbols or language contained in the dataset. Although the competition-driven nature of the tasks may foster faster progress toward agents that can better learn from human feedback, releasing all data and code may lead to researchers overfitting on the subset of tasks. This may result in a false signal of progress. There is therefore a need to utilize or develop more non-programmatic tasks in Minecraft [3] and other domains.

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

| Task | Question | Factor |
|---|---|---|
| FindCave | **Quantitative** | |
| | Did this player find and enter a cave? | Found Cave |
| | **Qualitative** | |
| | Which player found a cave the fastest? (If neither found a cave, that is a draw) | Found Cave Faster |
| | Which player moved more quickly and efficiently? | More Quick and Efficient Movement |
| | Which player was better at looking for caves in areas they hadn't already explored? | Better Cave Search in Unknown Areas |
| | Which player was better at going to areas where it is more likely to find caves? | Better Navigation to Cave Areas |
| | Which player was better at noticing potential caves that entered its field of vision? | Better Cave Detection |
| | Which player was better at realizing when it had successfully found a cave? (In other words, which player was better at properly ending the minigame once it had entered a cave?) | Better Cave Perception |
| | Which player seemed more human-like (rather than a bot or computer player)? | More Human Like |
| MakeWaterfall | **Quantitative** | |
| | Did this player create a waterfall? | Created Waterfall |
| | Did this player end the video while looking at a player-constructed waterfall? | Looked at Waterfall |
| | **Qualitative** | |
| | Which player moved more efficiently? | More Quick and Efficient Movement |
| | Which player chose a better location for their waterfall? (If neither player created a waterfall, select "Draw".) | Better Location for Waterfall |
| | Which player took a better "picture" of the waterfall? (If neither player took a picture of a player-constructed waterfall, select "Draw") | Better Picture of Waterfall |
| | Which player seemed more human-like (rather than a bot or computer player)? | More Human Like |
| BuildVillageHouse | **Quantitative** | |
| | Did the player harm the village? Examples include taking animals from existing pens, damaging existing houses or farms, and attacking villagers. | Least Village Harm |
| | **Qualitative** | |
| | Which player chose a better location for their house? | Better House Location |
| | Which player's structure seemed most like a house? | More House-Like Structure |
| | Which player was better at removing unnecessary blocks (or never placing unnecessary blocks)? | Intentional Block Placing |
| | Which player was better at using the appropriate type of blocks (i.e., the ones that are used in other houses in the village)? | Appropriate Block Placing |
| | Which player's house better matched the "style" of the village? | Better Style Matching |
| | Which player built the better-looking house? | More Attractive House |
| | Which player seemed more human-like (rather than a bot or computer player)? | More Human Like |
| AnimalPen | **Quantitative** | |
| | Did the player harm the village? Examples include taking animals from existing pens, damaging existing houses or farms, and attacking villagers. | Least Village Harm |
| | Did the player build an enclosed space (pen) from which animals could not escape? (For this question, it is okay if the pen did not contain animals). | Better Enclosed Space |
| | Did the player's pen contain at least two animals of the same type? | Correct Number of Animals |
| | Did the player's pen contain only one type of animal (if pen contained no animals, answer is no. Monsters and villagers are not counted)? | Correct Type of Animals |
| | Did the player's pen have at least one gate that would allow players to enter and exit (using a block to jump over fence does not count)? | Proper Exit |
| | Was the player's pen built next to a house? | Next to House |
| | **Qualitative** | |
| | Which player chose a better location for their pen in the village? | Better Location |
| | Which player searched more effectively for animals to pen? | More Effective Search |
| | After finding the animals, which player penned them more effectively? | More Effective Penning |
| | Which player seemed more human-like (rather than a bot or computer player)? | More Human Like |

Table 10: Mapping for each task from question asked of human evaluators to presented attribute.