# OpenReview forum: "BEDD: The MineRL BASALT Evaluation and Demonstrations Dataset for Training and Benchmarking Agents that Solve Fuzzy Tasks"
_NeurIPS.cc/2023/Track/Datasets_and_Benchmarks — NeurIPS 2023 Datasets and Benchmarks Oral_

### Official Review · Reviewer_PcC9 · 2023-07-22
**BEDD: The MineRL BASALT Evaluation and Demonstrations Dataset for Training and Benchmarking Agents that Solve Fuzzy Tasks**

**Rating:** 8
**Confidence:** 3
**Clarity:** The paper is easy to follow and well-…

**Strengths:**

Overall on all these axes I believe this is a strong submission which will be of use to the community. One caveat to my review is that I do not directly work on Minecraft RL tasks or current RL interaction datasets.


1. The main data contributions (video data for behavioral cloning purposes) are highly relevant to the community.
2. The quality of the analysis of these successful examples goes beyond simple data scraping, with useful human annotations for success and failure cases. The authors appear to have put significant effort into data cleanliness and verification.
3. The evaluation benchmark and automated evaluation suggestions also seem very useful for quick iteration, as human eval is hard to set up. I did not have a chance to directly evaluate the code for this however.
4. The standardization of BASALT benchmarks through explicit evaluation criterion is likely useful for the community. Especially for ensuring that certain benchmarks stay rigorous.
5. Ambiguous tasks with unclear termination criteria are also useful to demonstrate complex behavior. This paper suggested an interesting way to evaluate these methods using direct comparison (TrueSkill, ELO, etc.).

**Additional Feedback:**

N/A. Stated everything I want to say before. Overall excellent paper, I enjoyed reading it :-)

**Correctness:**

The dataset construction methodology is reasonably sound and thorough (see strengths).

**Documentation:**

The dataset is accessible. I was able to download the dataset and run a simple evaluation. Maintenance seems unnecessary, although it would be nice to add new examples as leaderboards progress. Documentation is good.

**Ethics:**

Any ethics concerns in this case are so abstract that they're not applicable.

**Limitations:**

The authors did provide a limitations section in the Appendix. Along with the suggestion above this limitations section seems reasonably thorough.

**Opportunities For Improvement:**

This paper would have been made stronger with annotated examples of failure cases, instead of only successful attempts. This would have provided interesting counterfactual information for people using the dataset. I see why the authors did not do this, as there are many degenerate failure possibilities. This could be a good avenue for future work (e.g. including information about "almost successes" in behavioral cloning methods).

More fine-grained annotation of successful episodes would also have been useful, including critical success keypoints (e.g. how important was a certain action for success?). However, annotation could always be more fine-grained so I consider this a minor complaint.

**Relation To Prior Work:**

This discussion was centered around past MineCraft datasets (e.g. BASALT and MineDojo). This background info seems reasonably thorough, however I'm not an expert in this area.

**Summary And Contributions:**

This paper builds on the Minecraft BASALT dataset by introducing:

1. New evaluation criteria to systematize metrics for methods
2. Data in the form of human and machine video/action examples which successfully complete certain tasks on MineCraft. They also include some in-depth analysis and annotation of these examples in the form of an evaluations dataset, which is human annotated.
3. Automated evaluation metrics to improve speed of iteration on BASALT tasks.

---

> ### Author Response · Authors · 2023-08-14
>
> Thank you for your enthusiastic review! We are pleased that you see the relevance of our data contributions to the community, as well as the effort and resulting quality of our data set.
>
> Thank you for your advice to analyze failure cases! In our submission, we took a closer look at our demonstration data. Due to our data collection pipeline, our demonstration dataset is high-quality, with less than 1% of idiosyncratic episodes. As a result, we do not have examples of almost successes in the demonstration dataset. Some examples of failures include players seemingly just playing Minecraft instead of accomplishing the goal or immediately dying (see Appendix C.2.2 for more details). These failure cases are perhaps less interesting, since they would also be captured by the MineRL Diamond dataset, However, these examples, as well as others in the MineDojo and MineRL Diamond datasets, could be used if one wants to include incorrect goals for a contrastive learning approach.
>
> However, based on your recommendation, we also took a closer look at the evaluation dataset, which consists of detailed human evaluations of both human and AI agents. We noticed a few interesting findings. In general, we found that the answers to the detailed evaluation questions provided by the human evaluators could be used as labels for different failure cases for training algorithms that explicitly account for negative examples. For example, Figure 4b (MakeWaterfall) in the paper reveals that, although Team GoUp’s algorithm can create waterfalls at a rate more similar to the human players, it struggles along all other criteria, including choosing a good location and taking a high-quality photograph. When decomposed individually, these details could be used as labels to describe a demonstration that creates a waterfall but does not choose a good location. As another example, Figure 4a (FindCave) suggests that, while Team GoUp’s algorithm still struggles to find caves, it can reasonably search for and navigate to areas that are likely to have caves. This finding suggests that the performance bottleneck may be the cave detection system employed by this approach. The videos generated by Team GoUp’s algorithm for FindCave could be used to distinguish between full and partial success on this task. We include these details in the updated version of the paper (Section 5.2 in orange).
>
> We also appreciate the suggestion to investigate successful episodes in more detail. We would love to understand what would be perceived as useful for a potential user of our dataset. We have included two potentially useful measures: a proxy for the distance traveled and a proxy for the number of blocks placed (details in Section 4). Because our demonstration dataset largely consists of successful episodes, summary statistics of these measures could be useful for measuring the progress of a new approach.
>
> Please let us know if our response here, along with the updated paper draft, has sufficiently addressed your comments!

---

> > ### Comment · Reviewer_PcC9 · 2023-08-29
> >
> > This addresses my concerns! I will up my score to an 8, I think this dataset will be very useful.

---

### Official Review · Reviewer_Vxj5 · 2023-07-31
**Dataset and Benchmark for MineRL BASALT competition**

**Rating:** 6
**Confidence:** 3
**Correctness:** The presentation is sound, with some …
**Clarity:** The paper is well structured and read…

**Strengths:**

- Remarkable data collection effort: 14k videos of human play, with annotations that also score human behavior aspects (eg human-like score, over 3000 annotations of various human and algorithmic agents provided by human evaluators), in addition to conventional reward based scores
- Relevant topic: providing benchmark for algorithms that learn from human feedback to cope with complex open world environment in Minecraft
- A streamlined codebase to make it easy for the potential users to evaluate a new model and to compare it to the existing leaderboard
- Aiming at open sourcing whole dataset and workflows around it

**Additional Feedback:**

The authors use NeurIPS 2022 template as visible in the footnote of the draft. Switching to NeurIPS 2023 one is recommended.

**Documentation:**

Datasheet for the dataset is provided in the supplementary, with some additional info in the source code from supplementary material.

**Ethics:**

No ethical concerns given here in my opinion.

**Limitations:**

It is not clear whether the scale of the collected data (26M image-text pairs from 14k videos) is sufficient to test larger scale models. Authors should elaborate whether this is the case and how does it compare to data amounts employed in MineDojo.

**Opportunities For Improvement:**

- Authors should clarify better in what sense the released dataset and benchmarking procedures will be openly available. It is mentioned in the text, but references to open source repos etc are missing. Authors do provide open source implementation in the supplementary material.

- Authors should discuss in more depth relation to MineDoJo and benchmarks introduced there. It should become more clear what are the main novelties of the proposed work to the elements already existing in MineDoJo

**Relation To Prior Work:**

I think relation to MineDojo should be worked out in the paper better than the very short text piece that I think does not provide the reader information how current work is complementary to what MineDojo already provides.

**Summary And Contributions:**

In their study, authors propose a dataset to enable easy training and evaluation of algorithms settled within Minecraft environment. BASALT Evaluation and Demonstrations Dataset (BEDD) is a formalized benchmark, which authors think of as a resource for algorithm development and performance assessment in MineRL Basalt competition. The authors describe datasets and tasks that make up the benchmark and point out that whole workflow is open and set up to be automatized for users who would like to evaluate their algorithm in MineRL Basalt. They demonstrate the utility of the proposed dataset and benchmark by evaluating several baseline algorithms. Authors observe that there is still a large room for improvement in learning from human feedback on open world environments, and suggest that inclusion of their open code for benchmarking and evaluating agents will encourage the development of more effective approaches.

Contributions of the presented work are as following:

- Composing BASALT Evaluation and Demonstrations Dataset (BEDD), to enable systematic and standardized assessment of algorithms that learn from human feedback data within MineRL Basalt competition. Annotations for the dataset contain descriptions related to human performance, eg human-like score, in addition to conventional performance reward based scores
- Demonstrating utility of the proposed dataset & benchmark by providing analysis of several algorithms.

---

> ### Author Response · Authors · 2023-08-14
>
> Thank you for your detailed review! We are pleased that you see the value of this work and acknowledge our detailed analysis and extensive data collection efforts.
>
> Before responding to the nice suggestions for improvement, we wanted to take a moment to clarify a few points. Maybe this is what you meant; we just want to clarify in case there was a misunderstanding. First, we want to clarify that the demonstration dataset consists of state-action pairs, not state-text pairs. One can view the instructions as a goal described in natural language to annotate the entire video, but we do not release per-state text descriptions. Second, we want to clarify that the human evaluation dataset is not conducted using the data from the demonstration dataset. In our paper, we evaluated a few algorithmic and human players, where the videos of human play were generated separately from the demonstration data. Third, we want to clarify that the challenge of the BASALT tasks is that they do not have corresponding reward functions and cannot be evaluated without the help of human evaluations. We include one task, ObtainDiamondShovel, with a concrete reward function to help researchers develop their methods.
>
> With that, we would now like to respond to the identified limitations and opportunities for improvement. First, we apologize for any confusion about the availability of the data and resources. We have shared our data openly in the following links (effective immediately), and added the same links to our paper. Our code is on Github at https://github.com/minerllabs/basalt-benchmark , and our evaluation dataset is already available at https://zenodo.org/record/8021960 . The demonstration dataset is available at https://github.com/openai/Video-Pre-Training#basalt-2022-dataset .
>
> Second, we appreciate the suggestion to better clarify the relationship of this work to MineDojo. We updated the paper with a more extensive comparison (orange text in Related Work). To summarize here, MineDojo aimed to provide a massive dataset covering a variety of possible tasks in Minecraft. Because this data stemmed from various online videos, it is quite diverse, which can prove challenging in settings where high-quality and consistent data is needed (such as training BC-based approaches). In contrast, our demonstration dataset was collected using verified, paid contractors who were instructed how they should play the game. For that reason, researchers can more readily use this data for techniques that rely on high-quality data for training, without the need for extensive data pre-processing and cleaning.
>
> Furthermore, MineDojo evaluates agents based only on binary success and failure conditions, whereas we present more extensive evaluation criteria with individual, decomposed evaluation criteria. We would be remiss to not mention the utility of this more extensive evaluation criteria. Decomposing the evaluation enables better quantitative understanding of the conditions in which certain algorithms fail and opportunities for improvement. For example, Figure 4b in the paper reveals that, although Team GoUp’s algorithm can create waterfalls at a rate more similar to the human players, it struggles along all other criteria, including choosing a good location and taking a high-quality photograph. Only looking at the binary success/failure condition for creating a waterfall, as in MineDojo, would ignore these important nuances in the algorithm’s behavior.
>
> Finally, we open source an end-to-end pipeline for using our curated demonstration data to train an agent and subsequently compare that agent using human feedback against the existing leaderboard. In this sense, our dataset complements MineDojo’s, as one can use MineDojo dataset to pretrain models, then use our demonstration dataset to fine-tune the models for BASALT tasks, and finally use our evaluation pipeline for a more extensive model evaluation.
>
> Third, we would like to address the question about the scale of the data for training large models. BASALT tasks take place in vanilla Minecraft, so any large models trained for this purpose could be used in the training pipeline. Our data can then be used for fine tuning these large models. We believe that the scale of the collected data is sufficient, as it was used by participants in the BASALT 2022 competition in various ways with larger models, such as creating embedding datasets with VPT.
>
> Fourth and finally, thanks for noticing the template! We updated it to the NeurIPS 2023 version.
>
> Please let us know if our response here, along with the improved and updated draft of the paper, has sufficiently addressed your comments!

---

### Official Review · Reviewer_328S · 2023-08-01
**A dataset for LfHF algorithm development and performance assessment**

**Rating:** 10
**Confidence:** 4
**Clarity:** Yes

**Strengths:**

1. A very strong data set with detailed and carefully design.
2.  A codebase for further study.

**Additional Feedback:**

Very nice work.

**Correctness:**

The authors are all vey experienced in this area and the dataset, codebase and the results have matched the practice.

**Documentation:**

Yes

**Ethics:**

No such issue

**Limitations:**

Yes,  the authors adequately addressed the limitations and potential negative societal impact of their work.

**Opportunities For Improvement:**

Maybe, the paper could provide more tasks and benchmark on the dataset.

**Relation To Prior Work:**

Yes, the background  and related work are very concise.

**Summary And Contributions:**

In this paper, the authors introduced BEDD dataset, which consists of a collection of 26 million image-action pairs from nearly 14,000 videos of human players completing the BASALT tasks in Minecraft. It also includes over 3,000 dense pairwise human evaluations of human and algorithmic agents.

These comparisons serve as a fixed, preliminary leaderboard for evaluating newly-developed algorithms. To enable this comparison, we present a streamlined codebase for benchmarking new algorithms against the leaderboard. In addition to presenting these datasets, we conduct a detailed analysis of the data from both datasets to guide algorithm development and evaluation.

---

> ### Author Response · Authors · 2023-08-14
>
> Thank you for your glowing feedback! We are happy to hear that our efforts in carefully designing and collecting the dataset are not unnoticed. Thank you again.

---

### Author Response · Authors · 2023-08-14

We thank the reviewers for their thoughtful comments and the effort spent reviewing our paper. We appreciate that all reviewers saw the value in our carefully-designed and collected dataset(s), as well as our streamlined and open-source codebase for training algorithms that learn from human feedback. We have responded to each review and revised our paper (with changes in orange). We hope that our responses and paper revisions have sufficiently addressed all comments.

---

### Decision · Program_Chairs · 2023-09-22

**Decision:**

Accept (Oral)

**Comment:**

The paper has received a consistent set of review comments, including two strong acceptance reviews and one weak acceptance review. All reviewers have praised the substantial contributions made in terms of data collection, baseline construction, and codebase. In light of these positive assessments, I believe the paper merits a clear 'accept' recommendation.